# U-Pb and Lu-Hf Record of Two Metamorphic Events from the Peixe Alkaline Suite, Brasilia Belt: Textural and Isotopic Complexity in Zircon

Marco Helenio Coelho [1], Luís Felipe Romero [2], Maria Virginia Alves Martins [2,3], Werlem Holanda [2], Marcelo Salomão [2], Guilherme Loriato Potratz [2], Armando Dias Tavares [1] and Mauro Cesar Geraldes [2,*]

[1] Institute of Physics Armando Dias Tavares, Rio de Janeiro State University, Av. São Francisco Xavier, 524 CEP, Maracanã, Rio de Janeiro 20550-013, Brazil; marco.helenio@gmail.com (M.H.C.); tavares.armandodias@gmail.com (A.D.T.)

[2] Lab-4037F, Faculty of Geology, Rio de Janeiro State University, Av. São Francisco Xavier, 524 CEP, Maracanã, Rio de Janeiro 20550-013, Brazil; romerolipe@gmail.com (L.F.R.); virginia.martins@ua.pt (M.V.A.M.); werlemholanda@gmail.com (W.H.); salomao.mss@gmail.com (M.S.); geoloriato@gmail.com (G.L.P.)

[3] GeoBioTEc Researh Unity, Aveiro University, Campus de Santiago, 3810-193 Aveiro, Portugal

* Correspondence: geraldes@uerj.br

**Abstract:** U-Pb and Lu-Hf isotopes, by inductively coupled plasma mass spectrometry and laser ablation (ICP-MS-LA), are reported in zircon grains from the Peixe Alkaline Suite. This unit comprises alkaline rocks such as syenites with nepheline, albite-oligoclase-biotite, and pegmatitic bodies. The zircon grain was imaged by cathodoluminescence (CL), which allowed the characterization of features within the crystal. These features comprise complex zone crosscuts, showing the existence of pulses that caused the intrusion of isotopically younger phases into the interior of the grain on a millimetric scale. The U-Pb results suggest a metamorphic event with Pb loss at $579 \pm 3$ Ma. They can be interpreted because of the collisional regional event of the Brasilia Orogen (Mara Rosa Orogeny). A second age grouping at $548 \pm 2.5$ Ma (MSWD = 8), obtained in areas with high luminescence fading laterally to oscillatory zoned domains with variations in the abundance of isotopes, is 33 Ma younger, demonstrating a rejuvenation of these areas through Pb loss. It is interpreted here as a second metamorphic event related to a collisional event (Santa Terezinha de Goiás arc). The Lu-Hf results for these areas indicate εHf values between −10 and −17, suggesting the existence of magmatic isotopic rework in a crustal environment.

**Keywords:** zircon U-Pb and Lu-Hf ages; isotopic rehomogenization; Brasilia Belt; Peixe Alkaline Suite

## 1. Introduction

Zircon is the most used mineral in U-Pb geochronology, mainly due to its wide distribution and diversity of rock types. In addition, zircon easily accepts the entry of U into its crystal lattice in substitution for Zr and, in contrast, zircon does not accept the entry of common Pb, which allows us to interpret that practically all the Pb present in zircon is radiogenic, resulting from the decay of U and Th [1–3]. The selection of zircon crystals without fractures, inclusions, or colorlessness has proven to be of great importance for the success of its application in the U-Pb method. This careful selection of clear grains allows for more consistent and accurate ages. Techniques such as the abrasion technique suggested by [4] have been shown to reduce the discordance of analytical results, probably due to the elimination of the grain edge characterized by the most significant loss of Pb.

Technological advances in mass spectrometry and new laboratory techniques [5–8] have allowed us to use other mineral phases, such as monazite, perovskite, titanite, baddeleyite, and rutile. Another important reason for using these minerals (in addition to zircon) is their valuable information regarding the age of magmatism, metamorphism, and mineralization. These possibilities arise because each mineral has the cooling temperature of the U-Th-Pb system.

U-Pb analysis by inductively coupled plasma mass spectrometry and laser abrasion (ICP-MS-LA) was first used in 1985 [9] and has been an essential analytical tool for geosciences ever since [5,10]. This technique has been developed chiefly by researchers in the earth sciences, mainly due to its application to geological samples, and many of the essential renovations in the method have been developed in universities by geology departments [11], for example the optimization of the wavelength of laser radiation, the optics of the laser beam, and the improvement in the camera of the sample holder to study minerals and rocks. Historically, this development had to wait for the construction of equipment such as multicollectors and magnetic separators, among others. With the current availability of such equipment, geologists are using ICP-MS-LA to take measurements of isotopic composition not only in individual crystals but also in parts of these minerals on the scale of a few tens of microns [12–16].

The objective of this work is to investigate the episodic loss of Pb from zircon during the metamorphic process. It is based on numerous examples of minerals that lose Pb in thermo-tectonic events, as [17,18] demonstrated. In these events, the heating of the mineral, which the passage of hydrothermal fluids may accompany, allows the displacement of Pb through the crystal lattice of the host mineral [19–22]. In these examples, the lower intercept detects episodic loss in the concord diagram, interpreted as the age of regional metamorphism. In this sense, the upper intercept [23] can be interpreted as the rock's crystallization age, and a later event would be responsible for the heating and loss of Pb in a metamorphic episode.

## 2. Geological Setting

The Peixe Alkaline Suite is in the northern part of the Brasilia Fold Belt, comprising metasedimentary and metavolcanic rocks deposited in the Mesoproterozoic with subsequent magmatism and deformation occurring in the Neoproterozoic [24,25]. The Peixe Alkaline Suite comprises rocks whose composition and features may be interpreted as formed before the main crustal shortening phase of the Basilica Fold Belt. Late-stage deformations under simple shear conditions affected the body, enveloping it within a regional dextral sigmoidal structure [26]. The intrusive complex is an elongated plutonic body in contact with metasedimentary rocks, forming halos of contact metamorphism with zoned hydrothermal alteration. Pegmatites and nepheline veins extend irregularly from the central body into the surrounding host rocks.

Regionally, the study area comprises Archean rocks from the Goiás Block [27], represented by two greenstone belts (Guarinos and Pilar de Goiás) covered by a Mesoproterozoic metasedimentary sequence (Figure 1). Different schists and paragneisses [25] constitute this metasedimentary basement of amphibolite facies, called the Ticunzal Formation. The rocks from the Ticunzal Formation outcrop to the SE of the study area, with Paleoproterozoic sources and deposition age defined by U-Pb ages between 1.47 and 1.57 Ga. Deformation and metamorphism occurred during the Brasiliano orogeny in the Neoproterozoic [24–27].

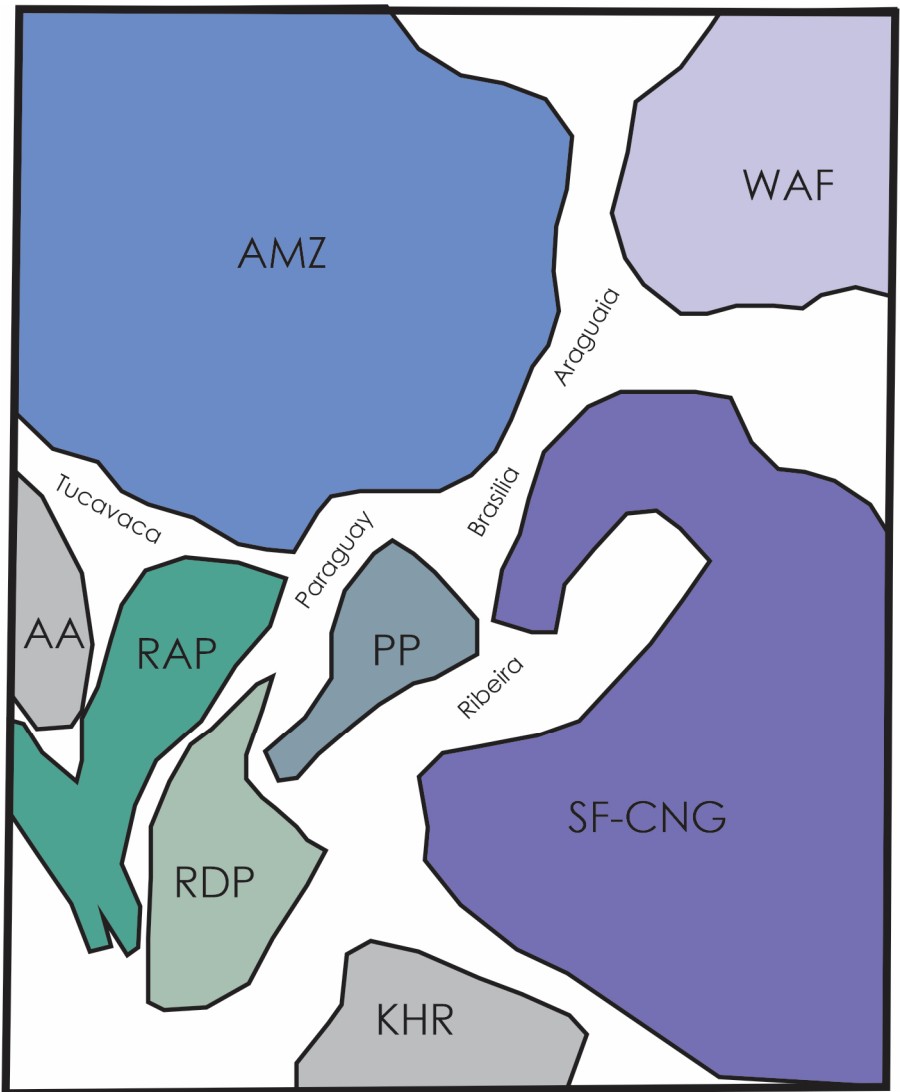

**Figure 1.** Gondwana assembly speculation. The collision period among each continental fragment is diachronic during the Gondwana amalgamation. Cratonic areas are AA—Arequipa-Antofala; AMZ—Amazonia; WAF—West Africa; RAP—Rio Apa; RDP—Rio de La Plata; PP–Papanapanema; SF—São Francisco; CNG—Congo; KHR—Kalahari. Modified from [28,29]. Tucavaca, Paraguay and Araguaia are Neoproterozoic mobile belts.

In this sense, the evolution of these basins and their sedimentation ages based on U-Pb geochronological data in detrital zircon obtained by the laser ablation method (MC-ICP-MS-LA) show an age of 1405 ± 10 Ma, marking the maximum limit of deposition and indicating a source area with Mesoproterozoic rocks. The samples show a higher frequency of Paleoproterozoic crystals between 1796 and 2472 Ma and older populations of Mesoneoarchean age between 2672 and 3112 Ma. In this sense, the Paleoproterozoic to Archean detrital zircon grains may come from the rocks that constitute the Block Archean of Goiás.

In the Neoproterozoic era, the Brasilia orogen developed, mainly represented by the calc-alkaline plutonic rocks of the magmatic arc that extend for more than 250 km and constitute the extension of the Goiás Magmatic Arc [26]. In addition, there are rocks from the Araguaia Orogen, whose rocks are pushed against the rocks of the Mara Rosa arc and older continental parts due to the collision of the Amazonian Craton. The platform sediments are deformed and pushed against its cratonic area, with ophiolitic rocks tectonically embedded in the Araguaia Belt. This metamorphic event probably results from the collisional processes

between the Amazonian craton and the São Francisco-Congo cratonic block (Figure 1). It comprises the late collision of the western Gondwana collage.

In the Brasilia Belt, in the states of Goiás and Tocantins, many pegmatites of syenitic composition are explored for gems. The Alkaline Peixe Suite hosts unusual mineral occurrences, including centimeter-sized zircon mega crystals that have been the subject of economic exploration. An important characterization was reported by [30,31]. Alkaline rocks defined the main units as being composed of nepheline syenites, albite-oligoclase-biotite sienite, and pegmatitic bodies mineralized with zircon, corundum, and rutile, and grouped these rocks under the denomination "Alkaline Monzonitic intrusives", including pegmatoid granites. Detailed studies and the first geological mapping of this unit were carried out by tourmaline, the term Peixe Alkaline Complex was adopted and petrochemical and geochronological studies were deepened with U-Pb zircon dating of a late pegmatite at ca. 550 Ma [32–38].

*2.1. Synthesis of the Rio do Peixe Alkaline Suite Geochronology from the Literature*

The authors of [38] reported LA-ICP-MS U-Pb ages in zircon from the Peixe Alkaline Suite. These authors present the results of two fragments of Peixe zircon. The 441 analyses result in concordant $^{206}Pb/^{238}U$ and $^{207}Pb/^{235}U$ mean ages of 571 ± 10 Ma (2%) and 568 ± 10 Ma (2%), respectively. The authors of [35] present the results of four fragments of zircon grains studied with the same protocol as [38] to complement previous experiments. The results (20 determinations per zircon grain) agree well with the initial data.

According to [31], geochronological analyses by U-Pb ICP-MS-LA on monazite from Peixe Alkaline Suite (locally named Boanerges pegmatite) generated an age of 519 ± 2.8 Ma. Chemical U-Th-Pb dating on uraninites from the São Júlio pegmatite revealed ages between 500 and 560 Ma, which are close to or overlap the age of ca. 560 Ma attributed to the leucogranites of the Mata Azul Suite in the literature [38]. However, the authors above suggest that there is a temporal distinction between the alkaline and acidic magmatic events found in the region.

The Pb/Pb ages in zircon, presented by [31], provided an age of 1470 ± 8 Ma and are similar to the U-Pb ages in syenite nepheline zircon. In addition, ref. [31] reported ages of 1503 ± 5 Ma in zircon grains from sienitic rocks, probably related to the Monte Santo Suite. However, ref. [30] found younger ages in zircon included in acid pegmatite, yielding 557 ± 15 Ma and 559 ± 7 Ma. These ages are interpreted as crystallization events.

Sm-Nd isotopic data in syenites show very heterogeneous TDM ages, ranging from Archean–Paleoproterozoic ages (1664–2979 Ma) with ℰNd(t = 1.5 Ga) from −7.22 to 8.34, and Mesoproterozoic TDM ages in a very short range (1395–1329 Ma), with ℰNd (t = 1.5 Ga) from −8.34 to 6.3 Ga [38]. The acid rocks, granites, and pegmatites reported by [29] to be associated with the alkaline system indicate Paleoproterozoic TDM between 2132 and 1919 Ma, all from crustal sources with ℰNd (t = 1.5 Ga) from −17.4 to −2.71.

In addition, U-Th-Pb ages on uraninite revealed a maximum age between 500 and 560 Ma, similar to the age of 560 Ma attributed to leucogranites of the Mata Azul Suite in the literature [38]. These ages, field relationships, mineralogy, and geochemical data suggest that the Mata Azul Suite is the likely source of the pegmatites that occur in the region and are termed the Mata Azul Suite Pegmatite Field. While incomplete U-Pb isotopic data are available in the literature, no published Lu-Hf data exist for the Peixe Alkaline Suite.

To summarize, the literature data do not allow a clear conclusion and a response to the question about the age of the Peixe Alkaline Suite. Is it Calymmian, around 1500 Ma, as reported by [31,38]? Alternatively, is it Ediacaran, around 570 Ma, as indicated at the beginning of the section? Or are the syenites Mesoproterozoic and the pegmatites Neoproterozoic? If the latter is the case, why are the syenites and the pegmatites considered

part of the same suite? In this paper, detailed U-Pb ages are presented, aiming to help understandings of the complex Peixe Alkaline Suite geologic evolution.

### 2.2. Local Context

The Peixe Alkaline Suite (Figure 2) comprises medium-grained foliated and banded leucocratic rocks represented by syenogranites (Figure 3A), nepheline syenites (Figure 3B), and alkaline pegmatites. These rocks exhibit recrystallized textures, with points in triple junctions, along with metamorphic albite and magnetite, indicatingthat they were subjected to low-grade metamorphism conditions [39]. Leucogranites (see map in Figure 4) were formed during the Neoproterozoic, associated with the evolution of the Mara Rosa Magmatic Arc located only in the north of Goiás. The essential minerals of these pegmatites are K-feldspar, quartz and mica, beryl, chlorite, garnet, albite, zwieselite, apatite, rockbridgeite, hagendorphyte, heterosite, phosphosiderite and strengite. Tourmaline-bearing pegmatites are also present, composed of tourmaline, albite, beryl, trilithionite, dravite, schorlite, elbaite, rossmanite, liddicoatite and dravite [37,38].

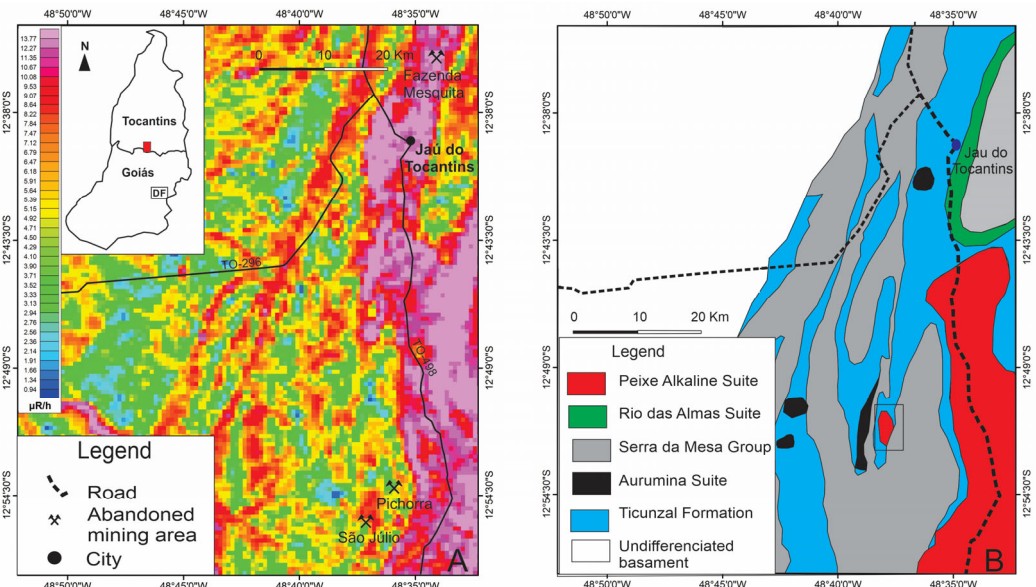

**Figure 2.** Geological context of the Peixe Alkaline Suite, in the context of the Brasilia Belt, in the state of Tocantins. Figure (**A**) (gamma spectrometry) highlights the alkaline intrusion and shows the area of gem exploration beyond the limits of the intrusion (in the host rocks). In (**B**), the modified geological map from [30] is on the same scale as that observed in (**A**). Black rectangle is the location of Figure 4.

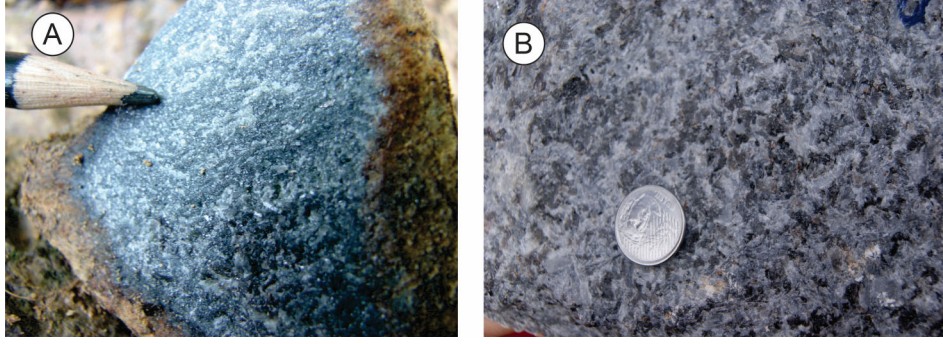

**Figure 3.** (**A**) Syenogranite sample of Peixe intrusion. (**B**) Nepheline syenite.

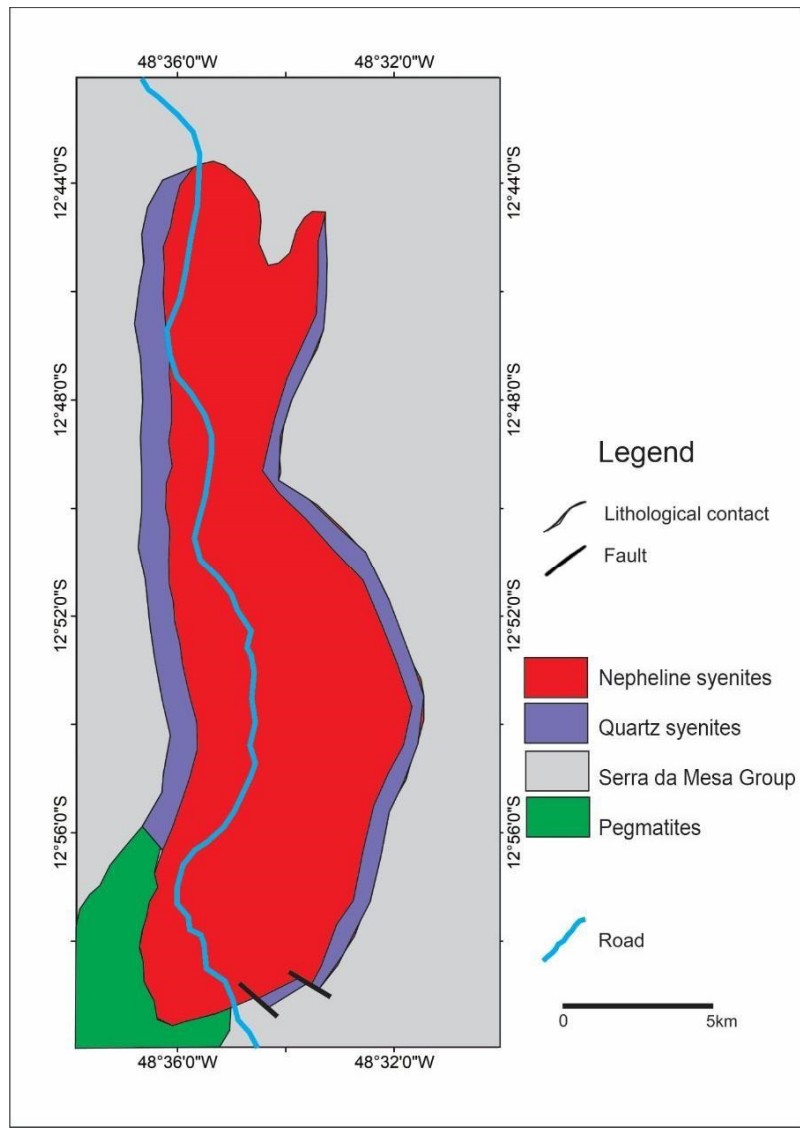

**Figure 4.** Local geological context of the Alkaline Peixe Suite in the region of economic exploitation through mining. Modified from [39,40].

The Peixe Alkaline Suite comprises elongated intrusions, controlled by a foliation concordant with the host rocks. The main outcropping rocks are nepheline syenite, alkali feldspar syenite, nepheline monzosyenite and nepheline monzodiorite with restricted areas of occurrence. Gneissic textures with banding observed are commonly observed at the outcrop scale. An important petrographic work describing the metamorphic textures in alkaline rocks is reported by [39,40]. In this work, the authors describe rocks of syenite composition that present nepheline as an abundant mineral and present typical miaskitic mineralogy, comprising albite, microcline, nepheline, amphibole, biotite, and magnetite and accessory clinopyroxene, calcite, sodalite, cancrinite, corundum, apatite, allanite, zircon and pyrochlore. Nepheline syenite presents fine to medium grains and hololeucocratic to leucocratic color. The predominant texture of the felsic minerals is granoblastic, where the crystals often form 120° triple junctions with medium grains and local domains, with a well-preserved igneous granular texture, although some rocks contain large crystals of albite, microcline, nepheline or perthite, commonly exhibiting recrystallization, characterized by feldspar crystal rims. Mafic minerals generally occur as oriented clusters with a cumulus texture, where amphibole and less commonly biotite occur as intercumulus minerals.

Rocks of monzosyenitic composition are medium- to coarse-grained and hololeuco-cratic, consisting almost entirely of feldspars and nepheline. The most common texture within these rocks is a zoned microstructure represented by cores of perthite, individual feldspars, or nepheline overlain by fine-grained granoblastic crystals of the same minerals. The alkali feldspar syenites are fine-grained, leucocratic, and foliated, and their contacts with nepheline syenite are through alkali feldspar–nepheline syenite transition zones. The dominant texture of the felsic minerals is granoblastic, while the mafic minerals are oriented biotite flakes and interstitial amphiboles. Large crystals of feldspar and perthite represent relict igneous textures, while small crystals of feldspar are granoblastic. Nepheline-free rocks contain only biotite and magnetite as mafic minerals, while nepheline-bearing rocks additionally contain amphibole and pyroxene. Monzodioritic rocks also contain nepheline and occur with foliation parallel to host rocks. These rocks are fine-grained and mesocratic, consisting of oligoclase, microcline, nepheline, clinopyroxene, amphibole, biotite, and accessory titanite and apatite, while fluorite and calcite occur as alteration minerals. Deformational textures are common in this rock, while the most commonly observed textures are large amphibole phenocrysts, with elongated grains in the direction of regional foliation.

## 3. Materials and Methods

### 3.1. Sample Preparation

Zircon grains were obtained from mines in the pegmatitic mineral gem production area hosted in metasyenitic rocks. Two kilograms of zircon grains was selected for this study (Figure 5). A large grain (2 cm × 3 cm × 1.5 cm) was fixed with adhesive tape in a circular mold with an internal diameter of 4 cm. The mold was filled with epoxy resin. The mounting face containing the zircon grain was polished after hardening. Aluminum oxide sandpaper with granulometry ranging from 127 to 15.3 μm was used in a sequence of abrasives of progressively smaller granulometry until half of the grain was exposed, allowing for the visualization of the grain's interior. After this procedure, diamond pastes with granulometry ranging from 3 to 0.25 μm were used until the resin and the grain were polished enough to be visualized with a scanning electron microscope (SEM).

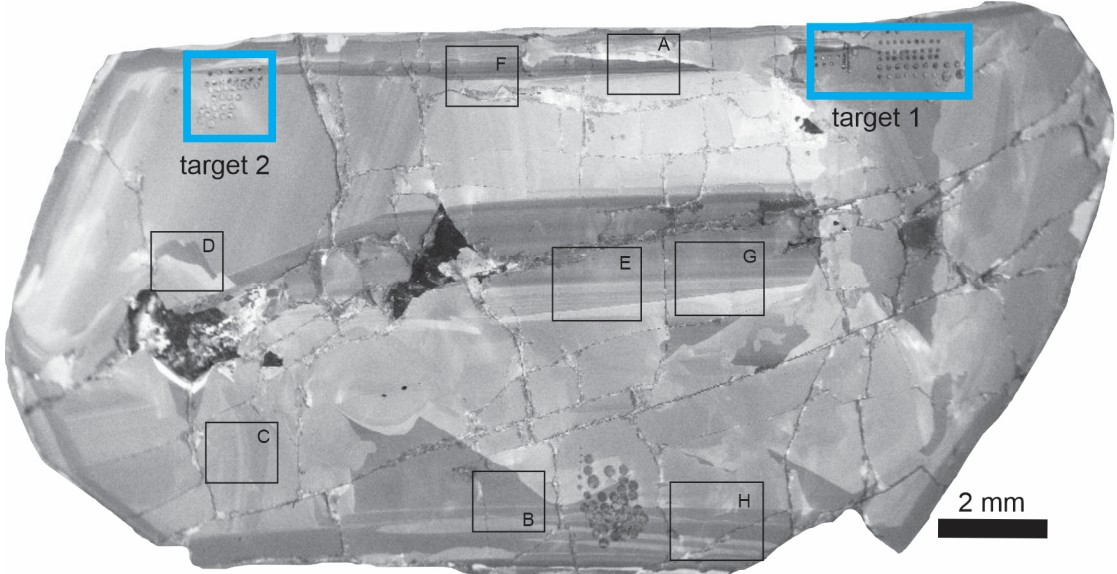

**Figure 5.** Cathodoluminescense (CL) image of the studied zircon grain. The rectangles represent details in Figure 6 (A, B, C, and D) and Figure 7 (E, F, G, and H). Target 1 and 2 present craters where the U-Pb and Lu-Hf analyses were carried out.

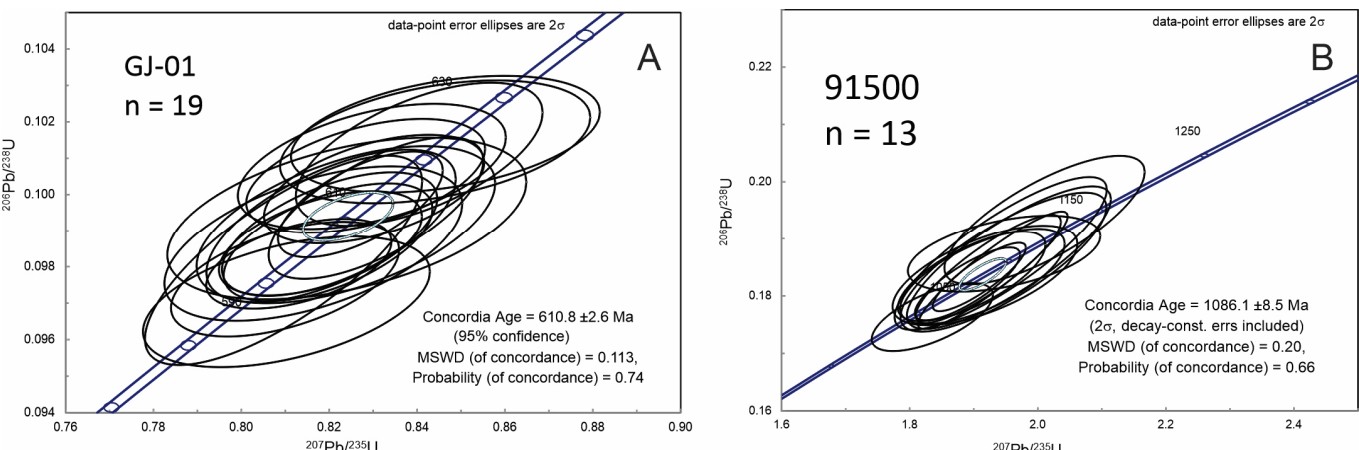

**Figure 6.** (**A**) Concordia diagram of GJ-1 (reference material) U-Pb isotopic results obtained in the laboratory during the analysis of unknown samples reported here. (**B**) Concordia diagram of 91500 (reference material).

### 3.2. Scanning Electron Microscopy

The centimetric zircon grain was imaged to evaluate its internal structure (Figure 5) using a cathodoluminescence CITL Mk5-2 coupled to the Scanning Electron Microscope (SEM) FEI Quanta 250 at the MultiLab facilities at Rio de Janeiro State University, with a voltage of 20 kV.

### 3.3. U-Pb Geochronological Analyses by ICP-MS-LA

U-Pb analyses were performed at the Multiuser Laboratory of Environment and Materials, MultiLab/Rio de Janeiro State University, using a Thermo Finnigan NEPTUNE PLUS MC-ICP-MS-LA coupled to a Laser Ablation Photon Machine ANALYTE G2 system with a 193 nm laser. Ablations were carried out using a 40 μm diameter spot. GJ-1 (Appendix A), 91500 (Appendix B), and zircon standards were used as reference materials. The following analytical sequence was followed: blank, GJ-1, 10 ablations of the unknown grain, reference materials GJ-1 (presented in Appendix A, Figure 6A), 91500 (presented in Appendix B, Figure 6B), and blank again. Concordia diagrams for age determination by the U-Pb method were produced with the Isoplot software version 4.15 [23]. The laser's wavelength of 193 nm produces a fine distribution of particles, which increases the efficiency of transporting the material, resulting in better sensitivity and minimal deposition in the plasma. A laser fluence of approximately 6 J/cm$^2$, a repetition rate of 5–7 Hz, and a spot size of 20–40 μm were used for all analyses. Data on the GJ-1 zircon standard for a 30 μm ablation spot usually yielded 432.000–114.000 cps of $^{206}$Pb, 25.000–7.000 cps of $^{207}$Pb, 6.500–4.200 cps of $^{208}$Pb, 4.400–4.200 cps of $^{202}$Hg and 1.060–1.090 cps of 204 (Hg + Pb). For $^{232}$Th and $^{238}$U measurements on Faraday cups, the values are 0.78 mV and 6.06 mV, respectively, yielding an age of 610.8 ± 2.6 Ma. The GJ-01 reference material comprises many zircon crystals, approximately 1 cm in size, from African pegmatites with a crystallization age of 608.5 ± 0.4 Ma [41]. Using an Excel spreadsheet, offline corrections for blank, Hg interference and standard lead were performed. In addition, the obtained GJ-1 reference material values were compared with the literature values [41]. Hence, the U-Pb results obtained by ICP-MS-LA were treated in an offline spreadsheet for blank and GJ-1 correction.

A second reference material, the 91.500, represents a single zircon crystal from a syenite pegmatite from the Renfrew County mine, Canada, which crystallized at 1065 ± 6 Ma [42].

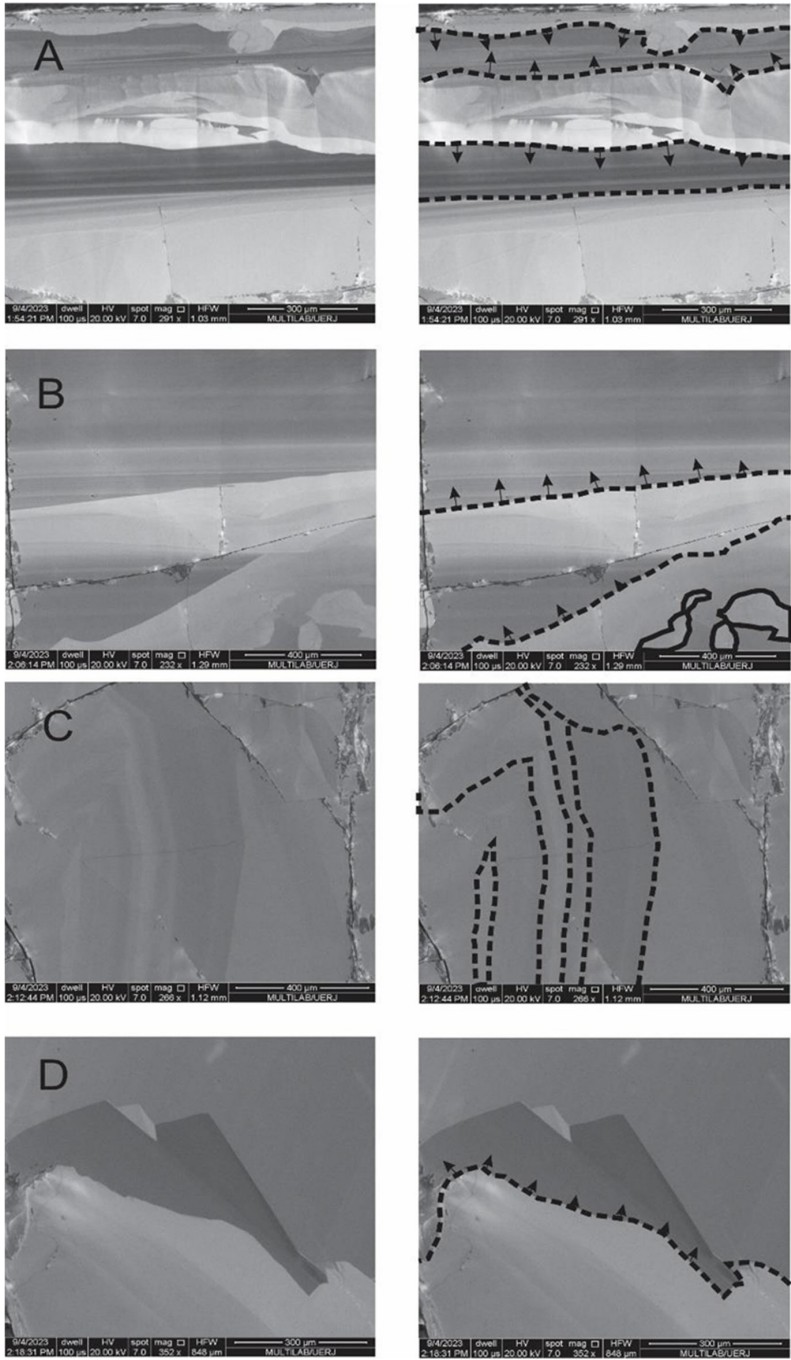

**Figure 7.** CL images of zircon grain showing textural complexity. See details in the text. Dash lines define the limit of metamorphic solution flux. Arrows are the direction of the metamorphic solutions. (**A**) Microlenses may represent new cycles of intrusion within the grain; (**B**) Sub-horizontal lenses display a cutting relationship; (**C**) Series of sub-parallel layers with variations in shades of light gray and dark gray material; (**D**) A relict portion of dark gray zircon surrounded by a lighter gray zircon portion in the upper part and light-toned areas that show growth over the relict portion in the lower part.

### 3.4. Lu-Hf Isotope Results

After the U/Pb isotopic analyses by the MC-ICP-MS-LA, zircon Hf isotope analysis was also carried out in situ at MultiLab (Rio de Janeiro State University, UERJ). Ref. [43] comprehensively described instrument conditions and data acquisition. CL images guided the isotopic analysis and were performed on zircon grains that show domains with distinct characteristics. The larger diameter of the laser-sampling crater during Lu-Hf analyses

may occasionally result in sampling different zircon growth domains or their mixture. This potential mix in the sample can cause scattering in the results, but this was avoided by choosing areas large enough to contain the respective craters (30 μm for U/Pb and 40 μm for Lu/Hf). Typical ablation times for cycles of each measurement were 50 s with a 7 Hz repetition rate and a laser power of 7 MJ/pulse, resulting in a beam depth of 30–40 μm. The carrier gases transported the ablated sample from the laser-ablation cell via a mixing chamber to the ICP-MS torch. The isobaric interference of $^{176}$Lu on $^{176}$Hf was corrected by measuring the intensity of the interference-free $^{175}$Lu isotope and using a recommended $^{176}$Lu/$^{175}$Lu ratio of 0.02655. Similarly, the isobaric interference of $^{176}$Yb on $^{176}$Hf was corrected by measuring the interference-free $^{172}$Yb isotope and using a recommended $^{176}$Yb/$^{172}$Yb ratio of 0.5886 [42] to calculate $^{176}$Hf/$^{177}$Hf values. Zircon 91500 was used as the reference material during our routine analyses, with a recommended $^{176}$Hf/$^{177}$Hf ratio of 0.282293 ± 28 from laser analyses. Initial $^{176}$Hf/$^{177}$Hf ratios were calculated concerning the chondritic reservoir at the time of zircon growth from magmas and using the U–Pb ages. εHf(t) values were determined to denote deviations in parts per 10,000 from a chondritic uniform reservoir (CHUR) reference in the initial $^{176}$Hf/$^{177}$Hf ratio between the sample and the CHUR reservoir at the time of zircon growth. We have adopted a decay constant for $^{176}$Lu of $1.867 \times 10^{-11}$ yr$^{-1}$, and the chondritic ratios of $^{176}$Hf/$^{177}$Hf (=0.282772) and $^{176}$Lu/$^{177}$Hf (=0.0332), as reported by [42]. Depleted-mantle model ages (TDM) were calculated using the measured $^{176}$Lu/$^{177}$Hf ratios, referred to as the depleted-mantle model with a present-day $^{176}$Hf/$^{177}$Hf = 0.28325, similar to that of mid-ocean ridge basalts (MORBs) and $^{176}$Lu/$^{177}$Hf = 0.0384 [42].

## 4. Results

### 4.1. CL Image of Zircon Grains

The CL images allowed the characterization of features within the zircon grain, such as fracture inclusions and growth phases. This information was used to select the area for ablation during the MC-ICP-MS-LA analysis. Advancements in analytical techniques using laser ablation have allowed a detailed investigation of isotopic variations within minerals. Consequently, knowledge of a mineral's internal composition has become crucial for robust interpretations of isotopic results in geochemical studies.

The studied Peixe Alkaline Complex zircon exhibits complex intersecting zones, indicating the incorporation of multiple pulses of younger material into its interior. In Figure 7A, microlenses may represent new cycles of intrusion within the grain, corresponding to geological events experienced by the studied rocks. Additionally, in Figure 7B, sub-horizontal lenses display a cutting relationship, suggesting that the lighter gray material intruded into the darker gray areas.

Figure 7C presents a series of sub-parallel layers with variations in shades of light gray and dark gray material, exhibiting curvilinear shapes and varying thicknesses. Fig-ure 7D shows a relict portion of dark gray zircon surrounded by a lighter gray zircon portion in the upper part and light-toned areas that show growth over the relict portion in the lower part.

Figure 8A shows lighter gray zircon lenses with curvilinear surfaces that cut through the darker areas. Figure 8B shows a lighter gray zircon that intersects darker gray portions at an angle; both zones display a potentially magmatic zoning pattern in Figure 8C (not confirmed by U-Pb ages obtained) and also show layers of dark gray zircon, probably of magmatic origin, cut by layers of lighter tones at an angle, suggesting a younger age for the lighter portion. Figure 8D shows a complex pattern characterized by amoeboid shapes of light gray zircon cutting through darker gray zircon. These sections are irregular with curvilinear contours, suggesting phases of fluid percolation within the initially darker gray zircon.

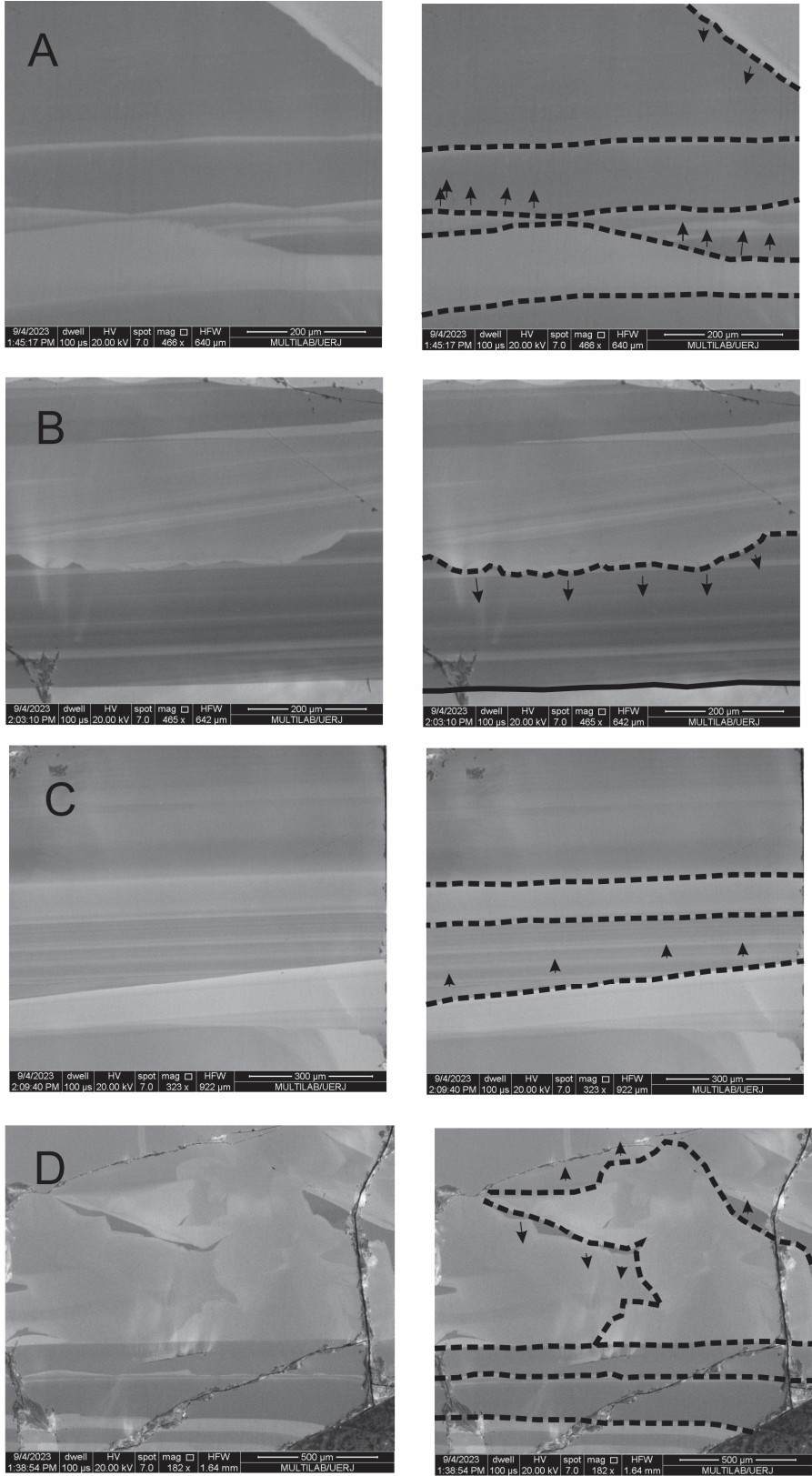

**Figure 8.** CL images of zircon grain showing textural complexity. See details in the text. Dash lines define the limit of metamorphic solution flux. Arrows are the direction of the metamorphic solutions. (**A**) Lighter gray zircon lenses with curvilinear surfaces that cut through the darker areas; (**B**) Lighter gray zircon that intersects darker gray portions at an angle; (**C**) Zones display a potentially magmatic zoning pattern and also show layers of dark gray zircon, probably of magmatic origin, cut by layers of lighter tones at an angle, suggesting a younger age for the lighter portion; (**D**) Complex pattern characterized by amoeboid shapes of light gray zircon cutting through darker gray zircon.

### 4.2. U-Pb and Lu-Hf Results in Mega Crystal Zircon of the Peixe Alkaline Suite

The U-Pb analysis was carried out in sectors of the zircon grain (Figure 3) highlighted in a blue rectangle with three targets. The prismatic habit zircon grain is 2 cm long and 1 cm wide with bipyramidal endings locally rounded. Fractures and inclusions are commonly seen. Eighteen spots (Table 1) were analyzed on the gray area exhibiting metamorphic zones, resulting in a Concordia age of 579 ± 3.1 Ma (MSWD = 2.2) interpreted as the metamorphic age of the zircon (Figure 9A). For this age group, twenty-one Lu-Hf analyses (Table 2) were carried out, yielding initial $^{176}Hf/^{177}Hf$ ratios from 0.28134 to 0.28275, $\varepsilon Hf(t)$ values between −8.6 and −13.6 (Figure 10A) and TDM model ages between 1.18 and 0.54 Ga. The complex feature observed in CL images shows that the zircon grain exhibits oscillatory zoning (Figure 11).

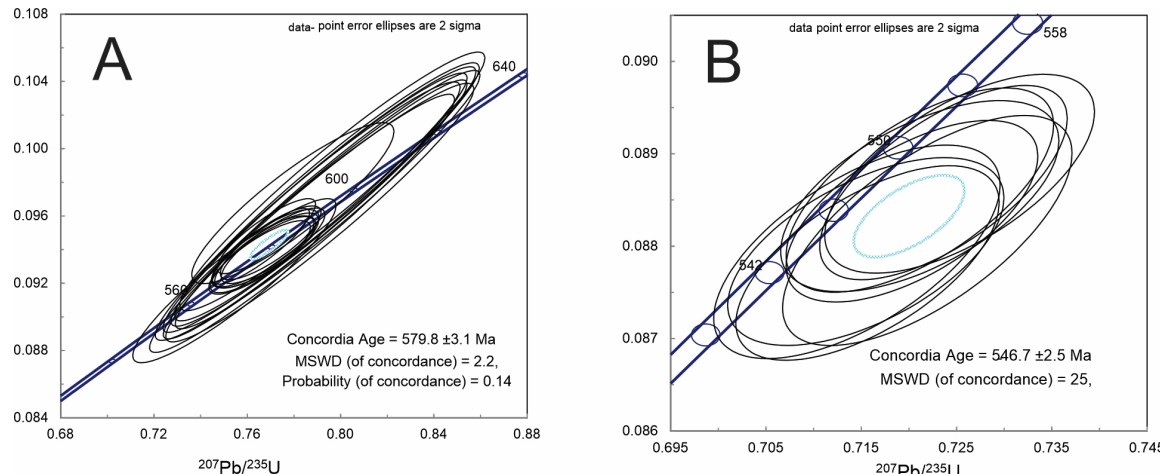

**Figure 9.** Concordia diagrams for the U-Pb results were obtained for two distinct areas of the analyzed grains: (**A**) Gray areas; (**B**) White areas.

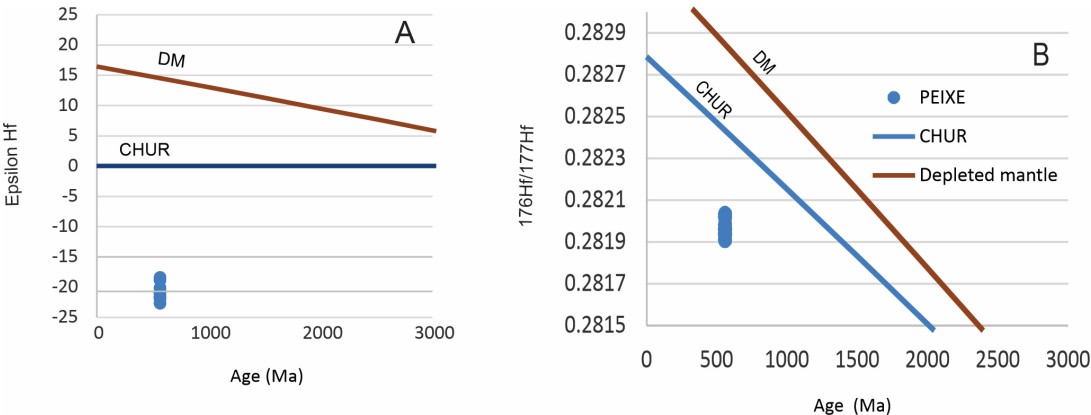

**Figure 10.** Lu-Hf results obtained on zircon grains from the Peixe Suite: (**A**) $\varepsilon Hf(t)$ values versus Age (Ma); (**B**) $^{176}Hf/^{177}Hf$ versus Age (Ma).

**Table 1.** U-Pb results of Peixe zircon gray area. Rho: relation between 207Pb/235U error and 206Pb/238U (define the ellipsis inclination). Conc.: Concordance between 207Pb/235U and 206Pb/238U ages. The f206: relation between common lead and radiogenic lead (based on 204Pb). Th/U: ratio between Th and U concentration.

| Peixe Zircon Gray Area Spot Number | | | | | Isotope Ratios | | | | | | | Age (Ma) | | Ages (Ma) | | Ages (Ma) | | |
|---|---|---|---|---|---|---|---|---|---|---|---|---|---|---|---|---|---|---|
| | Pb | Th | U | | 207Pb/ | 1 s | 206Pb/ | 1 s | | 207Pb/ | 1 s | 206Pb/ | 1 s | 207Pb/ | 1 s | 207Pb/ | 1 s | % |
| $f$ 206a | ppm | ppm | ppm | Th/Ub | 235U | [%] | 238U | [%] | Rho | 206P | [%] | 238U | abs | 235U | abs | 206Pb | abs | Concordance |
| PEIXE 01 01 | 0.00128 | 12.99 | 19.98 | 130.81 | 0.15 | 0.765874 | 2 | 0.094199 | 2 | 0.88 | 0.058967 | 1 | 580.3204 | 11 | 577.3934 | 13 | 565.8919 | 6 | 103 |
| PEIXE 01 02 | 0.001796 | 9.68 | 11.47 | 99.17 | 0.12 | 0.768137 | 2 | 0.094797 | 2 | 0.85 | 0.058768 | 1 | 583.8473 | 11 | 578.6935 | 13 | 558.5119 | 7 | 105 |
| PEIXE 01 03 | 0.002295 | 9.41 | 9.51 | 97.13 | 0.10 | 0.767691 | 2 | 0.094083 | 2 | 0.82 | 0.05918 | 1 | 579.6388 | 12 | 578.4372 | 14 | 573.7207 | 8 | 101 |
| PEIXE 01 04 | 0.001522 | 13.59 | 20.06 | 138.12 | 0.15 | 0.770034 | 2 | 0.094401 | 2 | 0.83 | 0.059161 | 1 | 581.5109 | 11 | 579.7827 | 13 | 573.0175 | 7 | 101 |
| PEIXE 01 05 | 0.001134 | 13.57 | 18.08 | 138.52 | 0.13 | 0.76638 | 2 | 0.093697 | 2 | 0.83 | 0.059322 | 1 | 577.3658 | 11 | 577.6842 | 14 | 578.9366 | 8 | 100 |
| PEIXE 01 06 | 0.001204 | 12.49 | 16.35 | 127.61 | 0.13 | 0.771073 | 2 | 0.094226 | 2 | 0.83 | 0.05935 | 1 | 580.4806 | 11 | 580.3781 | 14 | 579.9768 | 8 | 100 |
| PEIXE 01 07 | 0.00155 | 10.89 | 14.15 | 111.44 | 0.13 | 0.768319 | 2 | 0.093945 | 2 | 0.82 | 0.059315 | 1 | 578.8269 | 11 | 578.7981 | 14 | 578.6849 | 8 | 100 |
| PEIXE 01 08 | 0.001202 | 11.05 | 14.41 | 113.36 | 0.13 | 0.765302 | 2 | 0.093534 | 2 | 0.82 | 0.059342 | 1 | 576.4049 | 11 | 577.0646 | 13 | 579.6629 | 8 | 99 |
| PEIXE 01 09 | 0.002108 | 9.97 | 12.98 | 102.25 | 0.13 | 0.766982 | 2 | 0.093867 | 2 | 0.78 | 0.059261 | 1 | 578.3648 | 11 | 578.0303 | 14 | 576.7154 | 9 | 100 |
| PEIXE 02 01 | 0.002326 | 8.17 | 7.05 | 83.15 | 0.08 | 0.796508 | 7 | 0.098054 | 6 | 0.95 | 0.058914 | 2 | 602.9963 | 39 | 594.8567 | 40 | 563.9373 | 11 | 107 |
| PEIXE 02 02 | 0.001926 | 8.28 | 7.13 | 85.01 | 0.08 | 0.789097 | 7 | 0.096784 | 7 | 0.96 | 0.059132 | 2 | 595.5353 | 39 | 590.6596 | 40 | 571.971 | 10 | 104 |
| PEIXE 02 03 | 0.002426 | 8.58 | 7.22 | 88.01 | 0.08 | 0.792702 | 7 | 0.097158 | 6 | 0.96 | 0.059174 | 2 | 597.7337 | 39 | 592.7034 | 40 | 573.4936 | 11 | 104 |
| PEIXE 02 04 | 0.001543 | 8.75 | 7.15 | 89.92 | 0.08 | 0.794221 | 7 | 0.097435 | 6 | 0.96 | 0.059119 | 2 | 599.3599 | 39 | 593.5633 | 40 | 571.4688 | 11 | 105 |
| PEIXE 02 05 | 0.001796 | 9.20 | 7.72 | 94.94 | 0.08 | 0.789149 | 7 | 0.096387 | 7 | 0.97 | 0.05938 | 2 | 593.198 | 39 | 590.689 | 40 | 581.0615 | 10 | 102 |
| PEIXE 02 06 | 0.00249 | 9.52 | 8.58 | 97.83 | 0.09 | 0.791134 | 7 | 0.096199 | 7 | 0.96 | 0.059646 | 2 | 592.0952 | 39 | 591.815 | 40 | 590.7411 | 12 | 100 |
| PEIXE 02 07 | 0.002126 | 9.87 | 8.88 | 101.90 | 0.09 | 0.785935 | 7 | 0.096287 | 7 | 0.96 | 0.0592 | 2 | 592.6114 | 39 | 588.8632 | 40 | 574.4424 | 11 | 103 |
| PEIXE 02 08 | 0.002326 | 9.84 | 8.84 | 101.94 | 0.09 | 0.783532 | 7 | 0.095561 | 7 | 0.96 | 0.059467 | 2 | 588.3418 | 39 | 587.496 | 40 | 584.2294 | 11 | 101 |
| PEIXE 02 09 | 0.002625 | 9.40 | 8.35 | 96.72 | 0.09 | 0.794488 | 7 | 0.096972 | 6 | 0.96 | 0.059421 | 2 | 596.6382 | 39 | 593.7144 | 40 | 582.5577 | 11 | 102 |
| PEIXE 03 01 | 0.001581 | 8.37 | 8.29 | 85.09 | 0.10 | 0.779556 | 5 | 0.09676 | 4 | 0.89 | 0.058432 | 2 | 595.3944 | 24 | 585.2302 | 27 | 545.9909 | 11 | 109 |
| PEIXE 03 02 | 0.002829 | 9.03 | 9.46 | 97.01 | 0.10 | 0.754525 | 5 | 0.09215 | 4 | 0.92 | 0.059385 | 2 | 568.2373 | 25 | 570.8465 | 27 | 581.2516 | 11 | 98 |
| PEIXE 04 01 | 0.002108 | 9.64 | 9.24 | 100.22 | 0.09 | 0.835236 | 43 | 0.094699 | 9 | 0.22 | 0.063968 | 42 | 583.2703 | 54 | 616.5134 | 263 | 740.5632 | 308 | 79 |
| PEIXE 04 02 | 0.002326 | 8.92 | 8.95 | 94.34 | 0.09 | 0.822243 | 43 | 0.093251 | 9 | 0.22 | 0.063951 | 42 | 574.7344 | 54 | 609.2991 | 260 | 740.005 | 308 | 78 |
| PEIXE 04 03 | 0.001926 | 8.91 | 9.24 | 95.47 | 0.10 | 0.798328 | 44 | 0.09303 | 9 | 0.21 | 0.062238 | 43 | 573.4307 | 54 | 595.8847 | 261 | 682.3081 | 292 | 84 |
| PEIXE 04 04 | 0.002829 | 8.72 | 9.32 | 94.03 | 0.10 | 0.819438 | 42 | 0.092291 | 9 | 0.22 | 0.064396 | 41 | 569.0715 | 54 | 607.7351 | 258 | 754.6507 | 312 | 75 |
| PEIXE 04 05 | 0.002326 | 9.02 | 9.25 | 93.58 | 0.10 | 0.844645 | 43 | 0.095663 | 9 | 0.21 | 0.064037 | 42 | 588.9425 | 54 | 621.706 | 265 | 742.843 | 309 | 79 |
| PEIXE 04 06 | 0.002829 | 13.13 | 14.30 | 136.38 | 0.10 | 0.842121 | 43 | 0.095999 | 9 | 0.21 | 0.063622 | 42 | 590.9193 | 54 | 620.3153 | 266 | 729.0821 | 305 | 81 |
| PEIXE 04 07 | 0.002326 | 12.62 | 13.31 | 133.96 | 0.10 | 0.799636 | 44 | 0.094321 | 9 | 0.21 | 0.061487 | 43 | 581.0402 | 54 | 596.6233 | 264 | 656.3165 | 284 | 89 |
| PEIXE 04 08 | 0.002829 | 9.25 | 9.50 | 97.84 | 0.10 | 0.818945 | 43 | 0.095001 | 9 | 0.22 | 0.062521 | 43 | 585.0473 | 55 | 607.4594 | 265 | 691.9693 | 295 | 85 |
| PEIXE 04 09 | 0.002295 | 11.78 | 11.97 | 124.28 | 0.10 | 0.820754 | 43 | 0.093828 | 9 | 0.22 | 0.063442 | 42 | 578.1395 | 54 | 608.469 | 262 | 723.0823 | 304 | 80 |
| **Peixe zircon white area** | | | | | Isotope ratios | | | | | | | Age (Ma) | | Ages (Ma) | | Ages (Ma) | | |
| Spot number | Pb | Th | U | | 207Pb/ | 1 s | 206Pb/ | 1 s | | 207Pb/ | 1 s | 206Pb/ | 1 s | 207Pb/ | 1 s | 207Pb/ | 1 s | % |
| | $f$ 206 | ppm | ppm | ppm | Th/U | 235U | [%] | 238U | [%] | Rho | 206 | [%] | 238U | abs | 235U | abs | 206Pb | abs | Concordance |
| **Peixe zircon grain Area B** | | | | | | | | | | | | | | | | | | |
| PEIXE110_001 | 0.120321 | 21.72 | 6.144 | 0.050 | 0.305 | 0.72906 | 3 | 0.08801 | 2 | 0.61 | 0.06008 | 2 | 543.7572 | 9 | 606.4575 | 49 | 556.0014 | 12 | 98 |
| PEIXE110_004 | 0.008418 | 21.44 | 5.624 | 0.043 | 0.244 | 0.721924 | 3 | 0.08828 | 2 | 0.61 | 0.05931 | 2 | 545.3568 | 9 | 578.4941 | 48 | 551.8021 | 12 | 99 |
| PEIXE110_005 | 0.061995 | 21.73 | 5.710 | 0.041 | 0.235 | 0.724215 | 3 | 0.08811 | 2 | 0.61 | 0.059613 | 2 | 544.3497 | 9 | 589.5583 | 48 | 553.1524 | 12 | 98 |

**Table 1.** *Cont.*

| | | | | | | | | | | | | | | | | | | | |
|---|---|---|---|---|---|---|---|---|---|---|---|---|---|---|---|---|---|---|---|
| PEIXE110_005 | 0.015388 | 19.60 | 6.108 | 0.053 | 0.321 | 0.724666 | 3 | 0.08887 | 2 | 0.61 | 0.05914 | 2 | 548.8507 | 9 | 572.2537 | 49 | 553.4176 | 12 | 99 |
| PEIXE110_001 | 0.037668 | 18.53 | 5.180 | 0.066 | 0.339 | 0.728911 | 3 | 0.08999 | 2 | 0.62 | 0.058746 | 2 | 555.478 | 9 | 557.6993 | 48 | 555.9139 | 12 | 100 |
| PEIXE110_002 | 0.041611 | 19.88 | 5.612 | 0.058 | 0.326 | 0.715277 | 3 | 0.08912 | 2 | 0.62 | 0.05821 | 2 | 550.3306 | 9 | 537.6749 | 47 | 547.8747 | 12 | 100 |
| PEIXE110_003 | 0.147657 | 16.88 | 4.728 | 0.070 | 0.333 | 0.7264 | 3 | 0.08982 | 2 | 0.62 | 0.058655 | 2 | 554.4725 | 9 | 554.2971 | 49 | 554.4381 | 12 | 100 |
| PEIXE110_004 | 0.137561 | 20.09 | 5.728 | 0.054 | 0.307 | 0.714827 | 3 | 0.08824 | 2 | 0.62 | 0.058754 | 2 | 545.1198 | 9 | 557.9766 | 48 | 547.6085 | 12 | 100 |
| PEIXE110_005 | 0.07312 | 20.32 | 5.781 | 0.051 | 0.292 | 0.71738 | 3 | 0.08841 | 2 | 0.63 | 0.05885 | 2 | 546.1268 | 9 | 561.5514 | 47 | 549.1189 | 12 | 99 |
| PEIXE110_002 | 0.013023 | 24.97 | 7.181 | 0.026 | 0.184 | 0.720987 | 3 | 0.08656 | 2 | 0.62 | 0.06041 | 2 | 535.1603 | 9 | 618.292 | 47 | 551.2493 | 12 | 97 |
| PEIXE110_004 | 0.001138 | 25.84 | 7.420 | 0.027 | 0.197 | 0.726134 | 3 | 0.08669 | 2 | 0.61 | 0.06075 | 2 | 535.9315 | 9 | 630.3941 | 48 | 554.2814 | 12 | 97 |

**Table 2.** Lu-Hf results of Peixe zircon gray (A) and white (B) area. eHf(0): epsilon value for today. eHf(t): epsilon value for metamorphic age.

| Name | U/Pb Age (Ma) | ±2 s | Sample (Present Day Ratios) 176Hf/177Hf | ±2SE | 176Lu/177Hf | ±2SE | Chur 176Hf/177Hf (t) | DM 176Hf/177Hf (t) | Sample Initial Ratios 176Hf/177Hf (t) | eHf(0) | eHf(t) | ±2SE | T DM Crustal |
|---|---|---|---|---|---|---|---|---|---|---|---|---|---|
| **White Area** | | | | | | | | | | | | | |
| PEIXE_20 | 580 | 3 | 0.28204 | 0.000549 | 0.001326 | $9.99 \times 10^{-6}$ | 0.282323 | 0.282716 | 0.282021 | −26.357 | −10.6702 | 0.109901 | 2.128 |
| PEIXE_21 | 580 | 3 | 0.282004 | 0.0003 | 0.001367 | $1.16 \times 10^{-5}$ | 0.282323 | 0.282716 | 0.281985 | −27.6227 | −11.9577 | 0.124416 | 2.200 |
| PEIXE_22 | 580 | 3 | 0.282054 | 0.000231 | 0.001353 | $1.95 \times 10^{-5}$ | 0.282323 | 0.282716 | 0.282036 | −25.8391 | −10.1646 | 0.163178 | 2.100 |
| PEIXE_23 | 580 | 3 | 0.281983 | 0.000145 | 0.001199 | $1.95 \times 10^{-5}$ | 0.282323 | 0.282716 | 0.281966 | −28.3757 | −12.63 | 0.222312 | 2.238 |
| PEIXE_24 | 580 | 3 | 0.281982 | $8.29 \times 10^{-5}$ | 0.001187 | $1.57 \times 10^{-5}$ | 0.282323 | 0.282716 | 0.281966 | −28.388 | −12.6364 | 0.181189 | 2.238 |
| PEIXE_25 | 580 | 3 | 0.281976 | $6.64 \times 10^{-5}$ | 0.001175 | $1.16 \times 10^{-5}$ | 0.282323 | 0.282716 | 0.28196 | −28.6133 | −12.8562 | 0.140178 | 2.251 |
| PEIXE_26 | 580 | 3 | 0.281956 | $5.27 \times 10^{-5}$ | 0.00126 | $1.92 \times 10^{-5}$ | 0.282323 | 0.282716 | 0.281939 | −29.3126 | −13.5981 | 0.221017 | 2.292 |
| PEIXE_27 | 580 | 3 | 0.281955 | $4.68 \times 10^{-5}$ | 0.001261 | $1.68 \times 10^{-5}$ | 0.282323 | 0.282716 | 0.281937 | −29.3584 | −13.6446 | 0.195414 | 2.295 |
| PEIXE_28 | 580 | 3 | 0.281931 | $4.46 \times 10^{-5}$ | 0.00124 | $8.87 \times 10^{-7}$ | 0.282323 | 0.282716 | 0.281914 | −30.209 | −14.4863 | 0.024523 | 2.342 |
| PEIXE_29 | 580 | 3 | 0.281962 | $4.45 \times 10^{-5}$ | 0.001174 | $8.67 \times 10^{-6}$ | 0.282323 | 0.282716 | 0.281946 | −29.0887 | −13.332 | 0.111554 | 2.277 |
| PEIXE_30 | 580 | 3 | 0.282495 | $9.12 \times 10^{-5}$ | $8.84 \times 10^{-5}$ | $7.1 \times 10^{-7}$ | 0.282323 | 0.282716 | 0.282494 | −10.2395 | 6.077072 | 0.055704 | 1.182 |
| 91500 | 1065 | 10 | 0.282314 | 0.000104 | 0.000202 | $1.57 \times 10^{-6}$ | 0.28211 | 0.282471 | 0.28231 | −16.6418 | 7.093483 | 0.061696 | 1.391 |
| **Gray area** **Sample** | **U/Pb** **Age (Ma)** | **±2s** | **Sample (Present Day Ratios)** **176Hf/177Hf** | **±2 SE** | **176Lu/177Hf** | **±2SE** | **Chur** **176Hf/177Hf (t)** | **DM** **176Hf/177Hf (t)** | **Sample Initial Ratios** **176Hf/177Hf (t)** | **eHf(0)** | **eHf(t)** | **±2SE** | **Crustal** **residence** |
| 70 | 548 | 4 | 0.282038 | $5.55 \times 10^{-5}$ | 0.000891 | 0.000116 | 0.282429 | 0.282839 | 0.282029 | −26.4064 | −14.1802 | 1.947836 | 1793.539 |
| 80 | 548 | 4 | 0.282012 | $5.76 \times 10^{-5}$ | 0.000921 | 0.000115 | 0.282429 | 0.282839 | 0.282002 | −27.3274 | −15.1135 | 1.990984 | 1850.877 |
| 90 | 548 | 4 | 0.282061 | $7.24 \times 10^{-5}$ | 0.00099 | $9.96 \times 10^{-5}$ | 0.282429 | 0.282839 | 0.282051 | −25.5859 | −13.3958 | 1.447043 | 1745.301 |
| 100 | 548 | 4 | 0.281992 | $6.73 \times 10^{-5}$ | 0.000952 | 0.000132 | 0.282429 | 0.282839 | 0.281982 | −28.0286 | −15.8274 | 2.312122 | 1894.694 |
| 110 | 548 | 4 | 0.281982 | $7.62 \times 10^{-5}$ | 0.000899 | 0.000142 | 0.282429 | 0.282839 | 0.281973 | −28.388 | −16.1673 | 2.666763 | 1915.546 |
| 120 | 548 | 4 | 0.282021 | $8.41 \times 10^{-5}$ | 0.000852 | 0.000147 | 0.282429 | 0.282839 | 0.282012 | −27.011 | −14.7709 | 2.661478 | 1829.839 |
| 130 | 548 | 4 | 0.281918 | $7.45 \times 10^{-5}$ | 0.000859 | 0.00014 | 0.282429 | 0.282839 | 0.281909 | −30.6691 | −18.4363 | 3.15124 | 2054.514 |
| 140 | 548 | 4 | 0.28208 | $7.65 \times 10^{-5}$ | 0.000978 | 0.000113 | 0.282429 | 0.282839 | 0.28207 | −24.9139 | −12.7185 | 1.562188 | 1703.618 |
| 150 | 548 | 4 | 0.281961 | $7.55 \times 10^{-5}$ | 0.000952 | 0.000116 | 0.282429 | 0.282839 | 0.281951 | −29.1473 | −16.9474 | 2.190749 | 1963.362 |
| 160 | 548 | 4 | 0.282039 | $7.96 \times 10^{-5}$ | 0.001053 | $7.65 \times 10^{-5}$ | 0.282429 | 0.282839 | 0.282028 | −26.363 | −14.1975 | 1.136343 | 1794.605 |
| MUD TANK | 732 | 4 | 0.28246 | $4.32 \times 10^{-5}$ | $9.65 \times 10^{-5}$ | $3.5 \times 10^{-7}$ | 0.282323 | 0.282716 | 0.282458 | −11.5066 | 4.803988 | 0.044398 | 577.1016 |
| 91500 | 1065 | 4 | 0.282309 | $6.93 \times 10^{-5}$ | 0.000245 | $6.06 \times 10^{-7}$ | 0.28211 | 0.282471 | 0.282304 | −16.8477 | 6.856493 | 0.044412 | 374.8383 |

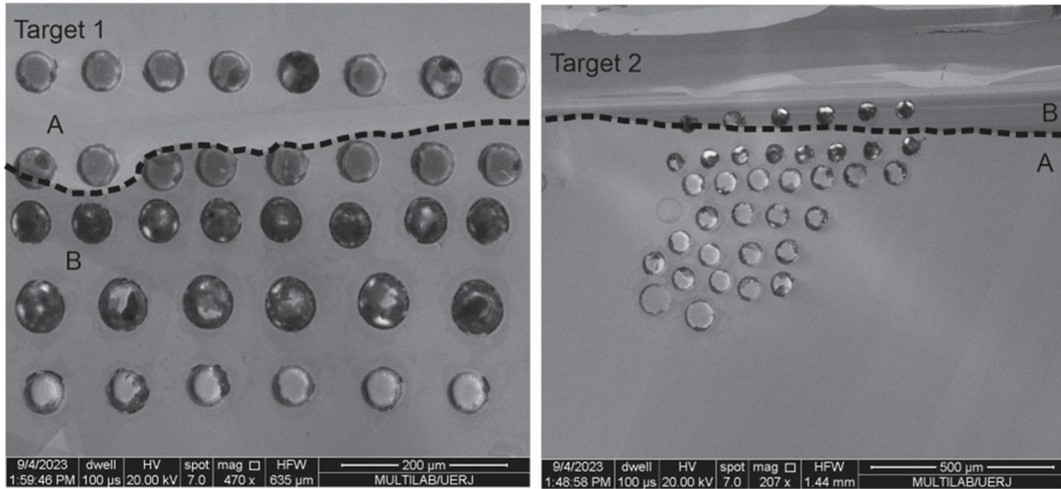

**Figure 11.** CL image of craters in two different areas of the zircon grain. See locations in Figure 5. Target 1 is comprised of area A; Target 2 is comprised of area B.

A second group of analyses was obtained in other zones (white, gray area; Figure 11) interpreted as younger domains of the zircon grains (light gray bands) (Table 1). Eleven analytical points give a range of concordant ages that spread out across the Concordia diagram (Figure 9B), giving a concordant age of 548 ± 2.5 Ma (MSWD = 8), with the 206Pb/238U average age of 521 ± 14 Ma (MSWD = 0.49). This age is interpreted as the best approximation of the metamorphic age of this area of the analyzed grain due to textural domains displaying different characteristics from the previous group (Table 2). Ten Lu-Hf analyses were performed in these domains, yielding initial $^{176}$Hf/$^{177}$Hf ratios from 0.28134 to 0.28275, ℰHf(t) values between −12.6 and −39.6 (Figure 10B), and TDM model ages between 1.28 and 0.64 Ga.

## 5. Discussions

### *5.1. The Tectonic Environment of Generation of the Peixe Alkaline Suite*

The intrusions related to the Peixe Alkaline Suite accompany the regional structures of the Brasilia Belt, intruding on the Mesoproterozoic metasedimentary sequences and suggesting an intrusion period related to the extensional process or rifting in the continental crust before the beginning of the seafloor expansion of the oceanic crust. These alkaline bodies showed Mesoproterozoic ages by the TIMS method on zircon, but the rocks metamorphosed in the Gondwana collision, as recorded by Neoproterozoic ages. In this way, the anorogenic magmatism of ca. 1.5 Ga studied here may relate to the Peixe syenite dated 1478 ± 8 Ma [29], interpreted by the authors as extensional rifting tectonic [25–27].

Magmatism accompanied the initial phase of continental rifting, leading to the rupture of the continental crust related to the extensional phase between 1.5 and 1.4 Ga, which marks the cratonization of Archean and Paleopoterozoic terrains observed regionally. Ref. [37] associated the massive intrusions with an aborted attempt at continental rifting. This anorogenic magmatism, associated with an alkali-rich extensional process in the state of Goiás, revealed two distinct episodes at approximately 1600 and 1770 Ma [44–46]. However, the ages reported in the literature may provide solid evidence of younger anorogenic magmatism. These dolerite dykes are known to be temporally and spatially associated with large ra-pakivi anorogenic granite plutons in other parts of the world, such as Finland [47], and are rich in Sn. These dolerite dykes are known to be temporally and spatially associated with large rapakivi anorogenic granite plutons in other parts of the world, such as Finland [45], and are rich in Sn. In this way, the anorogenic magmatism of ca. 1.5 Ga studied here may be related to the worldwide age-dependent process probably related

to supercontinent amalgamation, where Amazonian craton took part and the extensional setting responsible for the Peixe syenite origin dated to 1478 ± 8 Ma [29].

*5.2. The Record of Two Metamorphic Events (Pb Episodic Losses)*

Refs. [44,45] defined the crystallization age of the Alkaline Suite of Peixe at 1503 ± 3 Ma, consistent with preliminary U-Pb ages, obtained by the authors of [46], of 1470 ± 8 Ma. These Mesoproterozoic ages are consistent with the proposed geologic evolution [46]. Our results indicate that the zircon mega crystal obtained from the nepheline syenite rock of the Peixe Alkaline Complex underwent a metamorphic event at 579.7 ± 6.1 Ma, which can be interpreted because of the regional event related to the Brasilia Orogen. Our data are in good agreement with the results of [35], who reported the existence of a metamorphic–metasomatic event affecting the Peixe Alkaline Suite at around 577 Ma, and [38], who reported a mean $^{206}Pb/^{238}U$ age of 571 ± 10 Ma for a set of 441 concordant analyses.

A second event at 546.7 ± 5.3 Ma was identified in areas of the analyzed zircon grain characterized by distinct texture, as revealed by CL imaging and differences in elemental composition. This suggests a new event of isotopic homogenization within the studied zircon.

Two overlapping Neoproterozoic collisions represent the main magmatic phases of the Brasilia Orogen: the first is an island arc correlated with the accretion of the Mara Rosa arc, with U-Pb ages between 848 ± 4 Ma and 810 ± 4 Ma [37].

The εHf values between −17.8 and −13.9 obtained here for the 579.7 ± 6.1 Ma age domains indicate the existence of isotopic rehomogenization in a crustal environment, likely resulting from Neoproterozoic metamorphism associated with a collisional event during the formation of the supercontinent Gondwana.

Other authors [45,48,49] also suggested that the interpretation for the second group of younger ages obtained here could be explained by the existence of a second tectonic event called the Santa Terezinha de Goiás arc, formed by calc-alkaline magmatism. This magmatic event is recorded by U-Pb zircon ages ranging from 594 ± 2 Ma to 540 ± 5 Ma. The Lu-Hf results for these areas indicate εHf values between −10 and −17, suggesting the existence of isotopic rehomogenization in a crustal environment. These εHf values are partially similar to Sm-Nd isotope results, which indicate TDM ages between 2860 and 900 Ma and εNd (t = 580) from −22.42 to +2.84. The results reported here and data from the literature suggest that an essential episodic Pb loss event may be related to the crustal reworking resulting from collisions of cratonic fragments joined in the amalgamation of Gondwana.

## 6. Conclusions

The Alkaline Peixe Suite occurs in the central region of Brazil, in the Tocantins State, where many areas of exploration of pegmatites of syenitic composition can be observed. These rocks comprise the basement of the Brasilia Fold Belt, which intruded into metasedimentary and metavolcanic units deposited in the Mesoproterozoic [24,32,38,39]. The Pb/Pb ages in zircon presented by [33] provided an age of 1470 ± 8 Ma and are similar to the U-Pb ages in nepheline syenite zircon, obtained by [30], of 1503 ± 5 Ma.

The U-Pb Ages in zircon reported in the Peixe Alkaline Complex literature [29,30,32] present metamorphism results between 571 ± 10 Ma and 568 ± 10 Ma. The ages reported here in zircon grains show the concordant age of 579 ± 3.1 Ma (MSWD = 0.14) in part of the craters produced. This first event, indicated at 579 ± 3 Ma, can be interpreted as a result of the regional event related to the Brasilia Orogen. In contrast, a second age grouping was obtained, and other zones of the zircon grains were analyzed, giving a

concordant age of 548 ± 2.5 Ma (MSWD = 8). This Pb loss event was observed in the samples studied in this investigation, aged 33 Ma younger, and may represent a second regional metamorphism event.

The internal composition mapping of minerals has become crucial for robust interpretations of isotopic results in geochemical studies. The studied Peixe Alkaline Complex zircon exhibits complex intersecting zones, as CL images allowed the characterization of features within the zircon grain. The textures include fracture inclusions and growth, indicating the incorporation of multiple pulses of younger material into its interior.

Finally, Lu-Hf analyses were carried out in the areas of the grain previously analyzed by U-Pb with ƐHf values between −10 and −17, indicating a crustal source for the protoliths of the studied sample. These ƐHf values (T = 579 Ma) may result from a collisional event and the consequent metamorphism of the Peixe Alkaline Suite. Thus, the Peixe Alkaline Suite records two Pb loss events; the first is 33 My younger than the second one. These results are corroborated by the different isotope abundances measured by mass spectrometry and demonstrate a rehomoneization of these areas through Pb loss, interpreted here as a second metamorphic event during the Neoproterozoic, resulting from a collisional event in the formation of the supercontinent Gondwana.

**Author Contributions:** Conceptualization, M.H.C. and M.C.G.; methodology, L.F.R.; software, G.L.P.; validation, A.D.T., M.V.A.M. and L.F.R. formal analysis, M.H.C.; investigation, M.S.; resources, M.C.G.; Data curation, A.D.T.; writing—original draft preparation, M.H.C.; writing—review and editing, M.C.G.; visualization, W.H.; supervision, A.D.T.; project administration, M.V.A.M.; funding acquisition, M.C.G. All authors have read and agreed to the published version of the manuscript.

**Funding:** This research was funded by Fundação Carlos Chagas Filho de Amparo à Pesquisa do Estado do Rio de Janeiro for his postdoctoral scholarship, under process numbers E26-204.530/2021 and E26-204.531/2021, the Conselho Nacional de Desenvolvimento Científico e Tecnológico of Brazil (CnPQ) and Fundação Carlos Chagas Filho de Amparo à Pesquisa do Estado do Rio de Janeiro, Brazil, (FAPERJ) for the research grants (processes # 301470/2016-2 and E-26/202.843/2017, respectively), the Conselho Nacional de Desenvolvimento Científico e Tecnológico of Brazil, CnPQ (process #302676/2019-8) and FAPERJ (process #202.927/2019 and process #E—26/200.333/2023) for the research grants.

**Data Availability Statement:** Data are contained within the article.

**Acknowledgments:** The authors would like to thank the funding agencies: GLP thanks Fundação Carlos Chagas Filho de Amparo à Pesquisa do Estado do Rio de Janeiro for his postdoctoral scholarship; MCG would like to thank the Conselho Nacional de Desenvolvimento Científico e Tecnológico of Brazil (CnPQ) and Fundação Carlos Chagas Filho de Amparo à Pesquisa do Estado do Rio de Janeiro, Brazil, (FAPERJ) for the research grants; MVAM would like to thank the Conselho Nacional de Desenvolvimento Científico e Tecnológico of Brazil, CnPQ and FAPERJ for the research grants.

**Conflicts of Interest:** The authors declare no conflicts of interest.

## Appendix A. U-Pb Results of GJ-01 Reference Material

| Spot Number *f* 206a | Pb ppm | Th ppm | U ppm | Th/Ub | 207Pb/ 235U | 1 s [%] | 206Pb/ 238U | 1 s [%] | Rhod | 207Pb/ 206Pbe | 1 s [%] | 206Pb/ 238U | 1 s abs | 207Pb/ 235U | 1 s abs | 207Pb/ 206Pb | 1 s abs | % Concf |
|---|---|---|---|---|---|---|---|---|---|---|---|---|---|---|---|---|---|---|
| GJ-01 01 | 0.005477 | 30.06283 | 6.08167 | 301.9534 | 0.020141 | 0.824008 | 4.052433 | 0.099336 | 1.834998 | 0.452814 | 0.060162 | 3.61317 | 610.5164 | 11.20296 | 610.282 | 24.73127 | 609.412 | 22.01909 | 100.1812 |
| GJ-01 02 | 0.004145 | 31.3251 | 6.141736 | 316.2859 | 0.019418 | 0.823797 | 3.371622 | 0.099166 | 1.785388 | 0.529534 | 0.06025 | 2.86011 | 609.519 | 10.88228 | 610.1644 | 20.57244 | 612.5609 | 17.51991 | 99.50341 |
| GJ-01 03 | 0.005426 | 31.8749 | 6.458264 | 322.9167 | 0.02 | 0.816388 | 3.393371 | 0.098647 | 1.764413 | 0.519959 | 0.060022 | 2.898588 | 606.4754 | 10.70073 | 606.0314 | 20.56489 | 604.3717 | 17.51825 | 100.3481 |
| GJ-01 04 | 0.000724 | 33.49377 | 6.926545 | 339.2693 | 0.020416 | 0.819743 | 2.369111 | 0.098682 | 1.347151 | 0.568631 | 0.060247 | 1.948813 | 606.6795 | 8.172888 | 607.9049 | 14.40194 | 612.475 | 11.93599 | 99.05375 |
| GJ-01 05 | 0.001065 | 29.70623 | 5.673455 | 299.9333 | 0.018916 | 0.820427 | 2.826345 | 0.099131 | 1.734537 | 0.613703 | 0.060025 | 2.231504 | 609.315 | 10.56879 | 608.2866 | 17.19228 | 604.4581 | 13.48851 | 100.8035 |
| GJ-01 06 | 0.00217 | 31.6 | 6.3 | 319.6013 | 0.019712 | 0.820088 | 2.440273 | 0.098907 | 1.410144 | 0.577863 | 0.060136 | 1.991589 | 607.9974 | 8.573639 | 608.0977 | 14.83925 | 608.4716 | 12.11825 | 99.92206 |
| GJ-01 07 | 0.003769 | 29.03816 | 5.728595 | 289.501 | 0.019788 | 0.844114 | 3.417973 | 0.10145 | 1.359496 | 0.397749 | 0.060346 | 3.13597 | 622.9016 | 8.468324 | 621.4136 | 21.23975 | 615.9987 | 19.31754 | 101.1206 |
| GJ-01 08 | 0.003769 | 29.26287 | 5.70305 | 289.1738 | 0.019722 | 0.846452 | 3.393735 | 0.101661 | 1.28754 | 0.379387 | 0.060387 | 3.140012 | 624.138 | 8.036027 | 622.7 | 21.13279 | 617.4784 | 19.3889 | 101.0785 |
| GJ-01 09 | 0.00236 | 31.92863 | 6.417043 | 325.7854 | 0.019697 | 0.818001 | 2.307615 | 0.098653 | 1.192795 | 0.516895 | 0.060137 | 1.97543 | 606.5078 | 7.234397 | 606.9325 | 14.00566 | 608.5186 | 12.02086 | 99.66956 |
| GJ-01 10 | 0.002379 | 32.21277 | 6.405464 | 324.3235 | 0.01975 | 0.830222 | 2.46714 | 0.100147 | 1.220815 | 0.49483 | 0.060125 | 2.143919 | 615.2693 | 7.511301 | 613.7355 | 15.14171 | 608.0803 | 13.03675 | 101.1822 |
| GJ-01 11 | 0.002018 | 30.98723 | 6.194536 | 314.8791 | 0.019673 | 0.809951 | 2.619744 | 0.097666 | 1.321742 | 0.504531 | 0.060147 | 2.26187 | 600.7173 | 7.939933 | 602.4264 | 15.78203 | 608.8628 | 13.77169 | 98.66217 |
| GJ-01 12 | 0.003178 | 40.17856 | 7.972567 | 404.2939 | 0.01972 | 0.828294 | 2.682013 | 0.100513 | 1.275024 | 0.475398 | 0.059767 | 2.359557 | 617.4118 | 7.872147 | 612.665 | 16.43176 | 595.1566 | 14.04306 | 103.7394 |
| GJ-01 13 | 0.004987 | 24.32899 | 4.896935 | 248.996 | 0.019667 | 0.810259 | 3.292933 | 0.097041 | 1.510095 | 0.458587 | 0.060557 | 2.926264 | 597.0472 | 9.015981 | 602.599 | 19.84318 | 623.5398 | 18.24642 | 95.75125 |
| GJ-01 14 | 0.00414 | 38.87101 | 7.703065 | 390.2066 | 0.019741 | 0.829702 | 3.004319 | 0.100772 | 1.39789 | 0.465294 | 0.059715 | 2.659292 | 618.929 | 8.651947 | 613.4466 | 18.42989 | 593.259 | 15.77649 | 104.327 |

| Spot Number f 206a | Pb ppm | Th ppm | U ppm | Th/Ub | 207Pb/ 235U | 1 s [%] | 206Pb/ 238U | 1 s [%] | Rhod | 207Pb/ 206Pbe | 1 s [%] | 206Pb/ 238U | 1 s abs | 207Pb/ 235U | 1 s abs | 207Pb/ 206Pb | 1 s abs | % Concf |
|---|---|---|---|---|---|---|---|---|---|---|---|---|---|---|---|---|---|---|
| GJ-01 15 | 0.00265 | 38.45949 | 9.406425 | 385.8371 | 0.024379 | 0.830639 | 2.403451 | 0.099625 | 1.605276 | 0.667904 | 0.060471 | 1.788762 | 612.2084 | 9.827631 | 613.9668 | 14.75639 | 620.4576 | 11.09851 | 98.67046 |
| GJ-01 16 | 0.002694 | 24.74051 | 3.193575 | 253.3655 | 0.012605 | 0.809604 | 3.184728 | 0.098188 | 2.399141 | 0.753327 | 0.059801 | 2.094426 | 603.7836 | 14.48562 | 602.2317 | 19.17945 | 596.3944 | 12.49104 | 101.239 |
| GJ-01 17 | 0.002461 | 23.41418 | 4.962663 | 242.0717 | 0.020501 | 0.813095 | 2.119367 | 0.09809 | 1.031849 | 0.486866 | 0.06012 | 1.851217 | 603.204 | 6.224153 | 604.189 | 12.80498 | 607.8858 | 11.25329 | 99.22983 |
| GJ-01 18 | 0.001472 | 39.78582 | 7.637337 | 397.1309 | 0.019231 | 0.827085 | 1.852878 | 0.099723 | 1.196522 | 0.645764 | 0.060152 | 1.414741 | 612.7872 | 7.332136 | 611.9935 | 11.33949 | 609.0572 | 8.616581 | 100.6124 |
| GJ-01 19 | 0.003337 | 30.25869 | 5.503206 | 301.4032 | 0.018259 | 0.840478 | 2.730154 | 0.101107 | 1.591228 | 0.582834 | 0.06029 | 2.218499 | 620.8932 | 9.879825 | 619.4096 | 16.91084 | 613.9921 | 13.62141 | 101.124 |

## Appendix B. U-Pb Results of 91500 Reference Material

| Spot Number f 206a | Pb ppm | Th ppm | U ppm | Th/Ub | 207Pb/ 235U | 1 s [%] | 206Pb/ 238U | 1 s [%] | Rhod | 207Pb/ 206Pbe | 1 s [%] | 206Pb/ 238U | 1 s abs | 207Pb/ 235U | 1 s abs | 207Pb/ 206Pb | 1 s abs | % Concf |
|---|---|---|---|---|---|---|---|---|---|---|---|---|---|---|---|---|---|---|
| 91500 01 | 0.009961 | 8.792298 | 4.853542 | 46.54754 | 0.104271 | 1.835211 | 4.213146 | 0.176322 | 2.739343 | 0.650189 | 0.075488 | 3.201031 | 1046.848 | 28.67676 | 1058.147 | 44.5813 | 1081.53 | 34.62012 | 96.79323 |
| 91500 02 | 0.010988 | 8.596215 | −26.8882 | 42.96674 | −0.62579 | 1.947319 | 6.306179 | 0.1874 | 2.905876 | 0.460798 | 0.075364 | 5.596765 | 1107.277 | 32.17609 | 1097.523 | 69.21178 | 1078.233 | 60.34614 | 102.6937 |
| 91500 03 | 0.00739 | 8.071079 | 1.06961 | 39.96361 | 0.026765 | 1.890144 | 3.552354 | 0.1821 | 2.849466 | 0.802135 | 0.075281 | 2.121263 | 1078.436 | 30.72966 | 1077.633 | 38.28132 | 1076.009 | 22.82497 | 100.2255 |
| 91500 04 | 0.008758 | 8.399928 | 6.302696 | 40.96072 | 0.153872 | 1.913471 | 5.487691 | 0.183427 | 3.707488 | 0.675601 | 0.075659 | 4.045897 | 1085.668 | 40.251 | 1085.795 | 59.58507 | 1086.05 | 43.94047 | 99.96479 |
| 91500 05 | 0.021059 | 5.955597 | −7.68195 | 29.2964 | −0.26221 | 1.942041 | 6.969953 | 0.187465 | 5.828175 | 0.836186 | 0.075134 | 3.822646 | 1107.626 | 64.55439 | 1095.704 | 76.37003 | 1072.094 | 40.98237 | 103.3143 |
| 91500 06 | 0.009631 | 6.485514 | 6.235973 | 33.27502 | 0.187407 | 1.89662 | 4.96548 | 0.183421 | 3.476582 | 0.70015 | 0.074994 | 3.54533 | 1085.638 | 37.74311 | 1079.905 | 53.62247 | 1068.354 | 37.87666 | 101.6179 |
| 91500 07 | 0.018527 | 5.966631 | 118.018 | 29.74039 | 3.968275 | 1.946318 | 6.750591 | 0.185945 | 5.342696 | 0.791441 | 0.075915 | 4.126266 | 1099.372 | 58.73611 | 1097.178 | 74.06603 | 1092.83 | 45.09306 | 100.5987 |
| 91500 08 | 0.014414 | 6.582439 | 4.217703 | 31.60079 | 0.133468 | 2.010495 | 6.354589 | 0.193272 | 4.761473 | 0.749297 | 0.075445 | 4.208228 | 1139.076 | 54.2368 | 1119.058 | 71.11157 | 1080.392 | 45.46534 | 105.4318 |

| Spot Number<br>ƒ 206a | Pb<br>ppm | Th<br>ppm | U<br>ppm | Th/Ub | 207Pb/<br>235U | 1 s<br>[%] | 206Pb/<br>238U | 1 s<br>[%] | Rhod | 207Pb/<br>206Pbe | 1 s<br>[%] | 206Pb/<br>238U | 1 s<br>abs | 207Pb/<br>235U | 1 s<br>abs | 207Pb/<br>206Pb | 1 s<br>abs | %<br>Concf |
|---|---|---|---|---|---|---|---|---|---|---|---|---|---|---|---|---|---|---|
| 91500<br>09 | 0.013271 | 7.261859 | 4.198266 | 36.17678 | 0.116049 | 1.917528 | 6.151084 | 0.183603 | 3.87228 | 0.629528 | 0.075746 | 4.779255 | 1086.63 | 42.07737 | 1087.208 | 66.87507 | 1088.365 | 52.01576 | 99.84056 |
| 91500<br>10 | 0.019002 | 5.969732 | 3.916536 | 29.67938 | 0.131961 | 1.954757 | 6.659129 | 0.186735 | 5.45709 | 0.81949 | 0.075922 | 3.816302 | 1103.666 | 60.22802 | 1100.083 | 73.25593 | 1093.003 | 41.71229 | 100.9755 |
| 91500<br>11 | 0.002682 | 9.441448 | 19.663 | 44.92839 | 0.437652 | 1.922687 | 4.818485 | 0.18475 | 4.190815 | 0.869737 | 0.075479 | 2.377996 | 1092.87 | 45.80015 | 1089.002 | 52.47339 | 1081.275 | 25.71267 | 101.0723 |
| 91500<br>12 | 0.025914 | 4.625964 | 10.51357 | 21.27085 | 0.494271 | 2.104767 | 9.327857 | 0.199696 | 7.766245 | 0.832586 | 0.076442 | 5.166659 | 1173.689 | 91.15158 | 1150.367 | 107.3046 | 1106.671 | 57.17794 | 106.0558 |
| 91500<br>13 | 0.003693 | 11.4477 | 18.6438 | 55.76288 | 0.334341 | 1.929789 | 4.947299 | 0.184519 | 4.546394 | 0.918965 | 0.075852 | 1.950915 | 1091.617 | 49.62921 | 1091.466 | 53.99809 | 1091.165 | 21.2877 | 100.0414 |
| 91500<br>14 | 0.00283 | 10.546 | 22.49904 | 52.06992 | 0.432093 | 1.908628 | 3.115533 | 0.183301 | 2.658026 | 0.853153 | 0.075519 | 1.625253 | 1084.985 | 28.83917 | 1084.106 | 33.77567 | 1082.339 | 17.59075 | 100.2444 |

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
