# Peer review of "U-Pb and Lu-Hf Record of Two Metamorphic Events from the Peixe Alkaline Suite, Brasilia Belt: Textural and Isotopic Complexity in Zircon"

_minerals, doi:10.3390/min15030274_

Round 1

Reviewer 1 Report

Comments and Suggestions for Authors
  1. The Introduction section is too long and does not clearly present the scientific problem and the significance of this work.
  2. Data on reference materials such as 91500 can be moved to the appendix to prevent the paper from becoming overly long.
  3. The third section, "Economic exploration of pegmatite," is not necessary and is recommended to be shortened and merged into the second section and the introduction. The discussion on the controversy of formation age in section 3.1 could be included in the introduction.
  4. In line 203, "kg" should be changed to "kilogram," and in line 204, "cn" should be corrected to "cm."
  5. In this paper, Lu-Hf is not a geochronological method. Therefore, the term "geochronological" in line 257 should be removed.
  6. Why are the vertical axes different in the two plots in Figure 10?
  7. There are two "B" labels in Figure 5!
  8. The Pb content in Table 3 is not meaningful; what is meaningful is the common Pb content. The U content in the data for "Peixe zircon grain Area B" appears to be too low. Please check the data.
  9. In line 319, it is stated that there should be three targets in Figure 3, but only two targets are shown.
  10. In line 389, "81044"?
  11. In Figure 9A, the data appear to be divided into two groups. Do these two groups originate from different targets?

Author Response

Response to Reviwer 1

1. The Introduction section is too long and does not clearly present the scientific problem and the significance of this work. We shortened the Introduction and rewritten the objectives clearly.

  1. Data on reference materials (GJ-01 and 91500) were moved to appendix.

  1. We merged "Economic exploration of pegmatite," into  Geological setting.  We kepted the discussion on the controversy of formation age because it is important for results discussion.

  1. We changed "kg" to "kilogram. And "cn" to "cm."

5. We changed “Lu-Hf geochronological method” to “Lu-Hf Isotopes Results”.

  1. Figure 10 (A) vertical axes is Epsilon Hf and  Figure 10  (B) vertical axes is 176Hf/177  

  1. We corrected the "B" labels in Figure 5.

  1. We prefer to maintain the Pb concentration column because it may be used to correlate the older metamorphic area and younger metamorphic area. Common Pb may be observed in f206 column. Yes, the U content is really low.

  1. We corrected the Figure 5 using only two target areas.

  1. We corrected "81044" to “810’ Ma.

  1. The two groups are observed in the same area and present concordant age of 579 +/- 3.1 Ma

Reviewer 2 Report

Comments and Suggestions for Authors

Dear authors,

in the attached file some specific comments on your paper.

Pay attention please at the quality of the figures 2-6-9-10.

Author Response

Response to Reviwer 2

Suggestion

Line 14-15: Rewrite U-Pb and Lu-Hf isotopes by inductively coupled plasma mass spectrometry

and laser abrasion (ICP-MS-LA) are reported in zircon grains from the Peixe Alkaline Suite.

New text

U-Pb and Lu-Hf isotopes by inductively coupled plasma mass spectrometry and laser abrasion (ICP-MS-LA) are reported in zircon grains from the Peixe Alkaline Suite.

Suggestion

Line 17: cathodoluminescence (CL)

New text

cathodoluminescence (CL),

Suggestion

Line 21(Mara Rosa Orogeny?), why is reported a question mark?

New text

(Mara Rosa Orogeny).  

Suggestion

Line 22 Rewrite : “ A second age grouping at 548 ± 2.5 Ma (MSWD = 8), obtained in areas with

high luminescence fading laterally to oscillatory zoned domains with variations in the abundance of

isotopes, measured in the spectrometer is 31 Ma younger that demonstrated a rejuvenation of these areas through Pb loss.”

New text

A second age grouping at 548 ± 2.5 Ma (MSWD = 8), obtained in areas with high luminescence fading laterally to oscillatory zoned domains with variations in the abundance of isotopes, is 31 Ma younger that demonstrated a rejuvenation of these areas through Pb loss.

Suggestion

Line 65-72 Rewrite these lines to better explain which are the ages associated to the events of “regional metamorphism” and the “the upper intercept” that is reported in another paper ( reference 23) and how these data are related to the “episodic loss of Pb from zircon” of your paper.

New text

In the Brasilia Belt, in the states of Goiás and Tocantins, many pegmatites of syenitic composition are explored for gems. The Alkaline Peixe Suite hosts unusual mineral occurrences, including centimeter-sized zircon mega crystals that have been the subject of economic exploration.  An important characterization reported by [31; 30]. Alkaline rocks defined the main units as composed of nepheline syenites, albite-oligoclase-biotite sienite, and pegmatitic bodies mineralized with zircon, corundum, rutile, and  grouped these rocks under the denomination “Alkaline Monzonitic intrusives,” including pegmatoid granites. Detailed studies and the first geological mapping of this unit were carried out by tourmaline adopted the term Peixe Alkaline Complex and deepened the petrochemical and geochronological studies with U-Pb zircon dating of a late pegmatite at ca. 550 Ma [32; 33; 34; 35; 36; 37; 38].

Suggestion

Line 92-98: add references

New text

[32; 33; 34; 35; 36; 37; 38].

Suggestion

Line 101-107: add references

New text

Modified form [28] and [29].

Suggestion

Line 109-112: Caption of figure 1, the acronym AA and Ribeira are not defined.

New text

AA-Arequipa-Antofala

Suggestion

Line 119-127: Rewrite with the references reported in bracked at the end of the sentences.

New text

In this paper the presented detailed U-Pb ages aiming to  help the complex to understand  Peixe Alkaline Suite complex geologic evolution.

Suggestion

Line 141-146: Rewrite with the references reported in bracked at the end of the sentences and use

ICP-MS-LA

New text

Thermo Finningan NEPTUNE PLUS MC-ICP-MS-LA coupled to a Laser Ablation Photon Machine ANALYTE G2 system with a 193nm laser.

Suggestion

Line 147: Rewrite “U-Pb geochronological analyses by ICP-MS-LA

New text

U-Pb geochronological analyses by ICP-MS-LA

Suggestion

Line 159: TDM acronym

New text

Depleted-mantle model ages (TDM)

Suggestion

Line 167: delete ,

New text

Ok

Suggestion

Line 176: Add a sentence to highlight the contribute of the authors’ study “In this paper the presented study will help to understand…..” or similar

New text

In this paper the presented detailed U-Pb ages aiming to  help the complex to understand  Peixe Alkaline Suite complex geologic evolution.

Suggestion

Line 179-182: Verify the figures 3A-3B and 4 if are well cited

New text

The Peixe Alkaline Suite (Figure 2) comprises medium-grained foliated and banded leucocratic rocks represented by syenogranites (Figure 3A), nepheline syenites, and alkaline pegmatites. These rocks exhibit recrystallized textures, with points in triple junctions, along with metamorphic albite and magnetite, indicating (Figure 3B) that they were subjected to low-grade metamorphism conditions [39]. Leucogranites (Figure 4) were formed  during the Neoproterozoic, associated with the evolution of the Mara Rosa Magmatic Arc located only in the north of Goiás.

Suggestion

Line 204: 2cm

New text

(2cm x 3cm x 1.5 cm)

Suggestion

Line 213: Acronym CL images in figure 5

New text

Cathodoluminescense (CL)

Suggestion

Line 224: Write where is the laboratory

New text

Multiuser Laboratory of Environment and Materials - MultiLab/Rio de Janeiro State University

Suggestion

Line 248: use ICP-MS-LA

New text

MC-ICP-MS-LA

Suggestion

Line 258: use ICP-MS-LA

New text

MC-ICP-MS-LA

Line 276: eHf(t)

New text

εHf(t)

Suggestion

Line 280: )

New text

 176Hf/177Hf (= 0.282772)

Suggestion

Line 289: use ICP-MS-LA

MC-ICP-MS-LA

Suggestion

Line 300: Figure 6D is not present

New text

Figure 7D shows

Suggestion

Line 322: Figure 6A reports a concordia diagram

New text

Eighteen spots (Table 3) were analyzed on the gray area (Figure 10B) exhibiting metamorphic zones and resulted in a Concordia age of 579 ± 3.1 Ma (MSWD = 2.2) interpreted as the metamorphic age of the zircon (Figure 9A).

A second group of analyses was obtained in other zones (white, gray area; Figure 10) interpreted as younger domains of the zircon grains (light gray bands) (Table 3).  Eleven analytical points give a range of concordant ages that spread out across the Concordia diagram (Figure 9 B),

Suggestion

Line 325: Are twenty-five Lu-Hf analyses all reported in Table 4?

New text

twenty-one Lu-Hf analyses

Suggestion

Line 331: 206Pb/238U apex

Answer

We moved the Table 1 and Table 2 to appendix.

Suggestion

Line 348: Figure 11 is not discussed in the text

New text

Eighteen spots (Table 3) were analyzed on the gray area (Figure 11A) exhibiting metamorphic zones and resulted in a Concordia age of 579 ± 3.1 Ma (MSWD = 2.2) interpreted as the metamorphic age of the zircon (Figure 9A).

A second group of analyses was obtained in other zones (white, gray area; Figure 11B) interpreted as younger domains of the zircon grains (light gray bands) (Table 3).  Eleven analytical points give a range of concordant ages that spread out across the Concordia diagram (Figure 10 B),

Suggestion

Line 364: Use Sn as in the following line

New text

???

Suggestion

Line 372: results of the study are not discuss din this paragraph

New text

We discuss the regional rocks distribuition as showed in the geologic map as important results before the discussion of the U-Pb ages and Lu-Hf isotopes.

Suggestion

Line 403: Add references

New text

[25; 26; 27].

Suggestion

Line 418: in the abstract is reported 31 Ma not 33

New text

  A second age grouping at 548 ± 2.5 Ma (MSWD = 8), obtained in areas with high luminescence fading laterally to oscillatory zoned domains with variations in the abundance of isotopes, is 33 Ma younger that demonstrated a rejuvenation of these areas through Pb loss

Suggestion

Line 429: in the abstract is reported 31 Ma not 33

New text

  A second age grouping at 548 ± 2.5 Ma (MSWD = 8), obtained in areas with high luminescence fading laterally to oscillatory zoned domains with variations in the abundance of isotopes, is 33 Ma younger that demonstrated a rejuvenation of these areas through Pb loss.

Suggestion

REFERENCES

Kitajima, L. F. W., et al. "Uranium-lead ages of zircon megacrysts and zircon included in corundum

from Peixe Alkaline Complex (Brazil)." Simposio Sul Americano de Geologia de Isotopos. Vol. 3.

2001.

New text

Kitajima, L. F. W., et al. "Uranium-lead ages of zircon megacrysts and zircon included in corundum

from Peixe Alkaline Complex (Brazil)." Simposio Sul Americano de Geologia de Isotopos. Vol. 3.

2001.

Tables

Suggestion

In Tables 1, 2 and 3 report the standard error and the data with significative digits; explain Rho,

Concf and f206 a and Th/Ub; use apex for isotopic ratios.

Answer

We moved Tables 1 and Table 2 to appendix. Now theT ables 3 and 4  are (respectively) Tables 1 and 2.

Table 1. U-Pb results of Peixe zircon gray area. Rho (relation between 207Pb/235U  error and 206Pb/238U (define the ellipsis inclination). Conc. : Concordance between 207Pb/235U  and 206Pb/238U ages. The f206: relation between common lead and radiogenic lead (based on 204Pb); Th/U : ratio between Th and U concentration.

Suggestion

In Table 4 report the standard error and the data with significative digits; explain parameters eHf(0) and eHf(t), the data for CHUR, DM and TDM.

Answer

Table 2. Lu-Hf results of Peixe zircon gray (A) and white (B) area. eHf(0):  epsilon value for today. eHf(t): epsilon value for metamorphic age.

New text

Chondritic uniform reservoir (CHUR) reference in the initial 176Hf/177Hf ratio between the sample and the CHUR reservoir at the time of zircon growth. We have adopted a decay constant for 176Lu of 1.867 x 10-11 yr-1, and the chondritic ratios of 176Hf/177Hf (= 0.282772); and 176Lu/177Hf (=0.0332) as reported by [42]. Depleted-mantle model ages (TDM) were calculated using the measured 176Lu/177Hf ratios, referred to as the depleted-mantle model with a present-day 176Hf/177Hf=0.28325, similar to that of mid-ocean ridge basalts (MORB) and 176Lu/177Hf=0.0384 [42].

Reviewer 3 Report

Comments and Suggestions for Authors

The presented article evokes mixed feelings: 1) on the one hand, very interesting data with beautiful illustrations of zircon grains, 2) and on the other hand, the main conclusion about two metamorphic events is almost not supported by data on the mineralogy and petrography of the studied rocks. After all, it is not "zircon metamorphism" that is being studied, but the metamorphism of the rocks. There is only a mention of metamorphic albite and magnetite. But judging by the thin section (the only one), it is impossible to even assume that albite is of non-magmatic nature. Magnetite, by the way, is not even signed.

The question arises, how are the two distinguished metamorphic stages materially expressed? Which mineral assemblages were magmatic, and which were metamorphic? What are the PT-parameters of metamorphic events (if there was metamorphism)?

Without answers to these questions, it is difficult to agree that the article reveals two stages of metamorphism. The reviewer considers it necessary to supplement the article with these data. At the same time, there is redundancy in some figures or sections. Figures 2 and 4 can be combined. Section 3 "Economic exploration of pegmatite" is not directly related to the issue under discussion, it could be replaced by the section "mineralogical and petrographic features of the studied rocks".

Figure 1 does not show the scale and geographic coordinates. Even for such a schematic reconstruction, these elements should be.

Figure 2B contains a linear scale, but the dimension of the numerical scale (km) is not indicated.

Figure 3B - should be labeled "magnetite" - Mag.

Lines 388-389 "... the accretion of the Mara Rosa arc, with U-Pb ages between 848±4 Ma and 81044 ±4 Ma [37]." The age ranges should be corrected.

Comments on the Quality of English Language

Repetitions need to be reduced. For example, when citing uranium-thorium-lead chemical dating.

Author Response

Response to Reviwer 3

Suggestion

The presented article evokes mixed feelings: 1) on the one hand, very interesting data with beautiful illustrations of zircon grains, 2) and on the other hand, the main conclusion about two metamorphic events is almost not supported by data on the mineralogy and petrography of the studied rocks. After all, it is not "zircon metamorphism" that is being studied, but the metamorphism of the rocks. There is only a mention of metamorphic albite and magnetite. But judging by the thin section (the only one), it is impossible to even assume that albite is of non-magmatic nature. Magnetite, by the way, is not even signed.The question arises, how are the two distinguished metamorphic stages materially expressed? Which mineral assemblages were magmatic, and which were metamorphic? What are the PT-parameters of metamorphic events (if there was metamorphism)?

New text

The Peixe Alkaline Suite comprises elongated intrusions, controlled by a foliation concordant with the host rocks. The main outcropping rocks are nepheline syenite; alkali feldspar syenite, nepheline monzosyenite and nepheline monzodiorite with restricted areas of occurrence. Gneissic textures  with banding observed are commonly observed at outcrop scale. An important petrographic work describing the metamorphic textures in alkaline rocks is reported by [39]. In this work, the authors describe rocks of syenite composition that present nepheline as an abundant mineral and present typical miaskitic mineralogy, comprising albite, microcline, nepheline, amphibole, biotite, magnetite and accessory clinopyroxene, calcite, sodalite, cancrinite, corundum, apatite, allanite, zircon and pyrochlore. Nepheline syenite presents fine to medium grained and hololeucocratic to leucocratic grainedness. The predominant texture of the felsic minerals is granoblastic, where the crystals often forming 120º triple junctions with medium grained and local domains with well preserved igneous granular texture, although some rocks contain large crystals of albite, microcline, nepheline or perthite, commonly exhibiting recrystallization, characterized by feldspar crystal rims. Mafic minerals generally occur as oriented clusters with a cumulus texture, where amphibole and less commonly biotite occur as intercumulus minerals.

Rocks of monzosyenite composition are medium to coarse-grained and hololeucocratic, consisting almost entirely of feldspars and nepheline. The most common texture within these rocks is a zoned microstructure represented by cores of perthite, individual feldspars, or nepheline overlain by fine-grained granoblastic crystals of the same minerals. The alkali feldspar syenites are fine-grained, leucocratic, and foliated, and their contacts with nepheline syenite are through alkali feldspar-nepheline syenite transition zones. The dominant texture of the felsic minerals is granoblastic, while the mafic minerals are oriented biotite flakes and interstitial amphiboles. Large crystals of feldspar and perthite represent relict igneous textures, while small crystals of feldspar are granoblastic. Nepheline-free rocks contain only biotite and magnetite as mafic minerals, while nepheline-bearing rocks additionally contain amphibole and pyroxene. Monzodioritic rocks also contain nepheline and occur with foliation parallel to host-rocks. These rocks are fine-grained and mesocratic, consisting of oligoclase, microcline, nepheline, clinopyroxene, amphibole, biotite, and accessory titanite and apatite, while fluorite and calcite occur as alteration minerals. Deformational textures are common in this rock, while the most commonly observed textures are large amphibole phenocrysts, with elongated grains in the direction of regional foliation.

Suggestion

Without answers to these questions, it is difficult to agree that the article reveals two stages of metamorphism. The reviewer considers it necessary to supplement the article with these data. At the same time, there is redundancy in some figures or sections. Figures 2 and 4 can be combined.

 Answer

The authors prefere to keep Figure 2 and 4 separately because they present geologic maps of different scales.

Suggestion

Section 3 "Economic exploration of pegmatite" is not directly related to the issue under discussion, it could be replaced by the section "mineralogical and petrographic features of the studied rocks".

 Answer

We merged  "Economic exploration of pegmatite," into  Geological setting.  

Suggestion

Figure 1 does not show the scale and geographic coordinates. Even for such a schematic reconstruction, these elements should be.

Answer

Figure 1 is a Gondwana assembly speculation and usually are not presented with scale and coordinated

 Suggestion

Figure 2B contains a linear scale, but the dimension of the numerical scale (km) is not indicated.

Answer

We present a new map with numerical scale in km.

Suggestion

Figure 3B - should be labeled "magnetite" - Mag.

Answer

We included magnetite label in Figure 3B.

Suggestion

Lines 388-389 "... the accretion of the Mara Rosa arc, with U-Pb ages between 848±4 Ma and 81044 ±4 Ma [37]." The age ranges should be corrected.

Answer

We corrected "81044" to “810’ Ma.

 Suggestion

Repetitions need to be reduced. For example, when citing uranium-thorium-lead chemical dating.

New text

Chemical U-Th-Pb dating on uraninites from the São Júlio pegmatite revealed ages between 500 and 560 Ma, which are close to or overlap the age of ca. 560 Ma attributed to the leucogranites of the Mata Azul Suite in the literature [38]. However, the authors above suggest that there is a temporal distinction between the alkaline and acidic magmatic events found in the region.

Round 2

Reviewer 1 Report

Comments and Suggestions for Authors

The paper appears to have undergone significant improvements, but I believe the "Introduction" section still falls short and would benefit from a reorganization. This paper makes a new contribution to the chronology of the Peixe Alkaline Suite, rather than to zircon U-Pb geochronology. The Introduction should serve to introduce the scientific questions at hand, not the technical methodologies. The current manuscript clearly focuses on presenting the technical background, without mentioning the scientific problems to be addressed. Therefore, the Introduction section needs to be re-summarized.

Author Response

Response to Reviwer 1

  1. The Introduction section is too long and does not clearly present the scientific problem and the significance of this work. We shortened the Introduction and rewritten the objectives clearly.

  1. Data on reference materials (GJ-01 and 91500) were moved to appendix.

  1. We merged "Economic exploration of pegmatite," into  Geological setting.  We kepted the discussion on the controversy of formation age because it is important for results discussion.

  1. We changed "kg" to "kilogram. And "cn" to "cm."

  1. We changed “Lu-Hf geochronological method” to “Lu-Hf Isotopes Results”.

  1. Figure 10 (A) vertical axes is Epsilon Hf and  Figure 10  (B) vertical axes is 176Hf/177 

  1. We corrected the "B" labels in Figure 5.

  1. We prefer to maintain the Pb concentration column because it may be used to correlate the older metamorphic area and younger metamorphic area. Common Pb may be observed in f206 column. Yes, the U content is really low.

  1. We corrected the Figure 5 using only two target areas.

  1. We corrected "81044" to “810’ Ma.

  1. The two groups are observed in the same area and present concordant age of 579 +/- 3.1 Ma

Reviewer 3 Report

Comments and Suggestions for Authors

I accept your corrections. Good luck!

Author Response

(The authors gave the same response as above.)
